# An eQTL-based approach reveals candidate regulators of LINE-1 RNA levels in lymphoblastoid cells

**Juan I. Bravo**[1,2], **Chanelle R. Mizrahi**[1,3], **Seungsoo Kim**[4], **Lucia Zhang**[1,5], **Yousin Suh**[4,6], **Bérénice A. Benayoun**[1,7,8,9,10]*

1 Leonard Davis School of Gerontology, University of Southern California, Los Angeles, California, United States of America, 2 Graduate program in the Biology of Aging, University of Southern California, Los Angeles, California, United States of America, 3 USC Gerontology Enriching MSTEM to Enhance Diversity in Aging Program, University of Southern California, Los Angeles, California, United States of America, 4 Department of Obstetrics and Gynecology, Columbia University Irving Medical Center, New York, New York, United States of America, 5 Quantitative and Computational Biology Department, USC Dornsife College of Letters, Arts and Sciences, Los Angeles, California, United States of America, 6 Department of Genetics and Development, Columbia University Irving Medical Center, New York, New York, United States of America, 7 Molecular and Computational Biology Department, USC Dornsife College of Letters, Arts and Sciences, Los Angeles, California, United States of America, 8 Biochemistry and Molecular Medicine Department, USC Keck School of Medicine, Los Angeles, California, United States of America, 9 USC Norris Comprehensive Cancer Center, Epigenetics and Gene Regulation, Los Angeles, California, United States of America, 10 USC Stem Cell Initiative, Los Angeles, California, United States of America

* berenice.benayoun@usc.edu

**Data Availability Statement:** New sequencing data generated in this study is accessible through the Sequence Read Archive (SRA) under BioProject PRJNA937306. All code is available on the

## Abstract

Long interspersed element 1 (LINE-1; L1) are a family of transposons that occupy ~17% of the human genome. Though a small number of L1 copies remain capable of autonomous transposition, the overwhelming majority of copies are degenerate and immobile. Nevertheless, both mobile and immobile L1s can exert pleiotropic effects (promoting genome instability, inflammation, or cellular senescence) on their hosts, and L1's contributions to aging and aging diseases is an area of active research. However, because of the cell type-specific nature of transposon control, the catalogue of L1 regulators remains incomplete. Here, we employ an eQTL approach leveraging transcriptomic and genomic data from the GEUVADIS and 1000Genomes projects to computationally identify new candidate regulators of L1 RNA levels in lymphoblastoid cell lines. To cement the role of candidate genes in L1 regulation, we experimentally modulate the levels of top candidates in vitro, including *IL16*, *STARD5*, *HSD17B12*, and *RNF5*, and assess changes in TE family expression by Gene Set Enrichment Analysis (GSEA). Remarkably, we observe subtle but widespread upregulation of TE family expression following *IL16* and *STARD5* overexpression. Moreover, a short-term 24-hour exposure to recombinant human IL16 was sufficient to transiently induce subtle, but widespread, upregulation of *L1* subfamilies. Finally, we find that many L1 expression-associated genetic variants are co-associated with aging traits across genome-wide association study databases. Our results expand the catalogue of genes implicated in L1 RNA control and further suggest that L1-derived RNA contributes to aging processes. Given the ever-increasing availability of paired genomic and transcriptomic data, we anticipate this

Benayoun lab GitHub (https://github.com/BenayounLaboratory/TE-eQTL_LCLs). The Phase 3 autosomal SNVs generated by the 1000Genomes Project and called on the GRCh38 reference genome were obtained from The International Genome Sample Resource (IGSR) FTP site (http://ftp.1000genomes.ebi.ac.uk/vol1/ftp/data_collections/1000_genomes_project/release/20190312_biallelic_SNV_and_INDEL/). Structural variants were also obtained from the IGSR FTP site (http://ftp.1000genomes.ebi.ac.uk/vol1/ftp/phase3/integrated_sv_map/). mRNA-sequencing fastq files generated by the GEUVADIS consortium were obtained from ArrayExpress under accession E-GEUV-1.

**Funding:** This work was supported by National Science Foundation [https://www.nsf.gov/] Graduate Research Fellowship Program (NSF GRFP) DGE-1842487 (J.I.B.), National Institute on Aging [https://www.nia.nih.gov/] T32 AG052374 (J.I.B.), the University of Southern California with a Provost Fellowship (J.I.B.), National Institute on Aging [https://www.nia.nih.gov/] R25 AG076400 (C.R.M.), and National Institute of General Medical Sciences [https://www.nigms.nih.gov/] R35 GM142395 (to B.A.B). The funders had no role in study design, data collection and analysis, decision to publish, or preparation of the manuscript.

**Competing interests:** The authors have declared that no competing interests exist.

new approach to be a starting point for more comprehensive computational scans for regulators of transposon RNA levels.

## Author summary

Transposable elements, or jumping genes, are fragments of DNA that have or once had the ability to mobilize to a new location within our genome. In humans, the most abundant transposable element is LINE-1 (L1), accounting for ~17% of our total DNA. Though L1 is generally repressed in healthy human cells, derepression of transposable elements (including L1) has been observed in aging and in aging-associated diseases. Additionally, there is increasing evidence that L1 transcriptional levels may promote features of aging, highlighting the importance of understanding the mechanisms that regulate L1 RNA levels. Here, we computationally identify new candidate regulators of L1 RNA levels, provide experimental evidence that candidate regulators influence L1 RNA levels, and demonstrate that genetic variants associated with differences in L1 RNA levels are co-associated with aging phenotypes. Our approach expands the toolkit that can be used to characterize transposable element regulation and highlights specific genes for further study. Importantly, our results reiterate the notion that L1 levels are linked with aging phenotypes and represent a potential therapeutic target for age-related decline.

## Introduction

Transposable elements (TEs) constitute ~45% of the human genome [1]. Among these, the long interspersed element-1 (LINE-1 or L1) family of transposons is the most abundant, accounting for ~16–17% [1,2], and a tiny fraction remain autonomously mobile, with humans harboring an estimated 80–100 transposition-competent L1 copies [3]. These transposition-competent L1s belong to the evolutionarily younger L1Hs subfamily, are ~6 kilobases long, carry an internal promoter in their 5'-untranslated region (UTR), and encode two proteins—L1ORF1p and L1ORF2p —that are necessary for transposition [4]. In contrast, the remaining ~500,000 copies are non-autonomous or immobile because of the presence of inactivating mutations or truncations [1] and include L1 subfamilies of all evolutionary ages, including the evolutionarily older L1P and L1M subfamilies. Though not all copies are transposition competent, L1s can nevertheless contribute to aspects of aging [5,6] and aging-associated diseases [7–10].

Though mechanistic studies characterizing the role of L1 in aging and aging-conditions are limited, its effects are pleiotropic. For example, L1 can contribute to genome instability via insertional mutagenesis. Indeed, an expansion of L1 copy number with organismal aging [11] and during cellular senescence [12] have been documented, though many of these copies are likely cytosolic or extra-chromosomal. L1 can also play a contributing role in shaping inflammatory and cellular senescence phenotypes. The secretion of a panoply of pro-inflammatory factors is a hallmark of cell senescence, called the senescence associated secretory phenotype (SASP) [13]. Importantly, the SASP is believed to stimulate the innate immune system and contribute to chronic, low-grade, sterile inflammation with age, a phenomenon referred to as "inflamm-aging" [13,14]. During deep senescence, L1 are transcriptionally de-repressed and consequently generate cytosolic DNA that initiates an immune response consisting of the production and secretion of pro-inflammatory interferons [15]. Finally, L1 is causally implicated

in aging-associated diseases, including cancer. L1 may contribute to cancer by (i) serving as a source for chromosomal rearrangements that can lead to tumor-suppressor gene deletion [16] or (ii) introducing its active promoter next to normally silenced oncogenes [17]. Thus, because of the pathological effects L1 can have on hosts, it is critical that hosts maintain precise control over L1 activity.

Eukaryotic hosts have evolved several pre- and post-transcriptional mechanisms for regulating TEs [18,19]. Nevertheless, our knowledge of regulatory genes remains incomplete because of cell type-specific regulation and the complexity of methods required to identify regulators. Indeed, one clustered regularly interspaced short palindromic repeats (CRISPR) screen in two cancer cell lines for regulators of L1 transposition identified >150 genes involved in diverse biological functions (*e.g.* chromatin regulation, DNA replication, and DNA repair) [20]. However, only about ~36% of the genes identified in the primary screen exerted the same effects in both cell lines [20], highlighting the potentially cell type-specific nature of L1 control. Moreover, given the complexities of *in vitro* screens, especially in non-standard cell lines or primary cells, *in silico* screens for L1 regulators may facilitate the task of identifying and cataloguing candidate regulators across cell and tissue types. One such attempt was made by generating gene-TE co-expression networks from RNA sequencing (RNA-seq) data generated from multiple cancer-adjacent tissue types [21]. Although co-expression modules with known TE regulatory functions, such as interferon signaling, were correlated with TE modules, it is unclear whether other modules may harbor as of now uncharacterized TE-regulating properties, since no validation experiments were carried out. Additionally, this co-expression approach is limited, as no mechanistic directionality can be assigned between associated gene and TE clusters, complicating the prioritization of candidate regulatory genes for validation. Thus, there is a need for the incorporation of novel "omic" approaches to tackle this problem. Deciphering the machinery that controls TE activity in healthy somatic cells will be crucial, in order to identify checkpoints lost in diseased cells.

The 1000Genomes Project and GEUVADIS Consortium provide a rich set of genomic resources to explore the mechanisms of human TE regulation *in silico*. The 1000Genomes Project generated a huge collection of genomic data from thousands of human subjects across the world, including single nucleotide variant (SNV) and structural variant (SV) data [22,23]. To accomplish this, the project relied on lymphoblastoid cell lines (LCLs), which are generated by infecting resting B-cells in peripheral blood with Epstein-Barr virus (EBV). Several properties make them advantageous for use in large-scale projects (e.g. they can be generated relatively noninvasively, provide a means of obtaining an unlimited amount of a subject's DNA and other biomolecules, and can serve as an *in vitro* model for studying the effects of genetic variation on phenotypes of interest) [24,25]. Naturally, the GEUVADIS Consortium generated transcriptomic data for a subset of subjects sampled by the 1000Genomes Project and carried out an expression quantitative trait locus (eQTL) analysis to define the effects of genetic variation on gene expression [26]. Later, in a series of landmark studies on TE biology, this collection of data was reanalyzed (i) to characterize the effects of polymorphic TE structural variation on gene expression (TE-eQTLs) [27–29] and (ii) to explore the potential impact of TE polymorphisms on human health and disease through genome-wide association study (GWAS) analysis [30]. These results highlight the value of this data and the power of eQTL analysis in identifying genetic factors implicated in gene expression control and, potentially, disease susceptibility. Together, these resources provide a useful toolkit for investigating the genetic regulation of TEs, generally, and L1, specifically.

Much work on the mechanisms of L1 regulation has been carried out by looking exclusively at full-length, transposition-competent L1 elements, as this has allowed for the study of the whole L1 replication lifecycle, starting from transcription at the internal promoter and ending

with transposition into a new genomic site [20,31]. However, the total L1 RNA pools can be influenced by a number of other sources, including L1 copies residing in introns, L1 copies that are exonized, L1 copies that are co-transcribed because of nearby genes, and L1 copies that are independently transcribed from their own promoter regardless of their ability to mobilize. Thus, cellular L1 RNA levels are likely to be modulated by a number of transcriptional and post-transcriptional processes, including promoter-dependent transcription, RNA turnover, exonization, and/or intron retention (among others). However, the control mechanisms for non-full-length and transposition-incompetent L1 elements remain incompletely characterized, even though there is increasing evidence that these can have important regulatory and functional potential. For example, one study suggested that intronic L1s are part of an important regulatory network maintaining T-cell quiescence [32]. This is consistent with the increasing appreciation for the importance of alternative splicing in immune regulation and cell death pathways [33] and with another pair of studies highlighting the importance of TE exonization and intronic TE co-option in interferon signaling [34,35]. Additionally, L1 RNA may be sufficient to induce an interferon response and alter cellular viability, even in the absence of transposition [36,37]. Given these observations, there is a need to characterize the control mechanisms and functional effects of all L1 RNA sources, including both mobile and truncated, non-autonomous or transposition-incompetent copies.

In this study, we (i) develop a new pipeline to identify novel candidate regulators of L1 RNA levels in lymphoblastoid cell lines, and we apply it to both transposition-competent and -incompetent L1 subfamilies across genomic origins (intronic, intergenic, and exon-overlapping L1 RNA levels), (ii) provide experimental evidence for the involvement of top candidates in L1 RNA level control, and (iii) expand and reinforce the catalog of diseases linked to differential L1 levels.

## Results

### In silico scanning for L1 subfamily candidate regulators by eQTL analysis

To identify new candidate regulators of L1 RNA levels, we decided to leverage publicly available human "omic" datasets with both genetic and transcriptomic information. For this analysis, we focused on samples for which the following data was available: (i) mRNA-seq data from the GEUVADIS project, (ii) SNVs called from whole-genome sequencing data overlayed on the hg38 human reference genome made available by the 1000Genomes Project, and (iii) repeat structural variation data made available by the 1000Genomes Project. This yielded samples from 358 European and 86 Yoruban individuals, all of whom declared themselves to be healthy at the time of sample collection (Fig 1A). Using the GEUVADIS data, we obtained gene and TE subfamily expression counts using TEtranscripts [38]. As a quality control step, we checked whether mapping rates segregated with ancestry groups, which may bias results. However, the samples appeared to cluster by laboratory rather than by ancestry (S1A Fig). As additional quality control metrics, we also checked whether the SNV and SV data segregated by ancestry following principal component analysis (PCA). These analyses demonstrated that the top two and the top three principal components from the SNV and SV data, respectively, segregated ancestry groups (S1B and S1C Fig).

We then chose to do a three-part integration of the available "omic" data (Fig 1B). Since TEtranscripts quantifies total TE RNA levels aggregated at TE subfamily resolution and discards TE position information, we chose to carry out a *trans*-eQTL analysis [39] against global RNA levels of each L1 subfamily. We note here that we were interested in understanding the genetic basis for variation in global L1 subfamily RNA levels. Thus, we did not carry out genetic association analysis against single insertion events, but against global RNA levels of

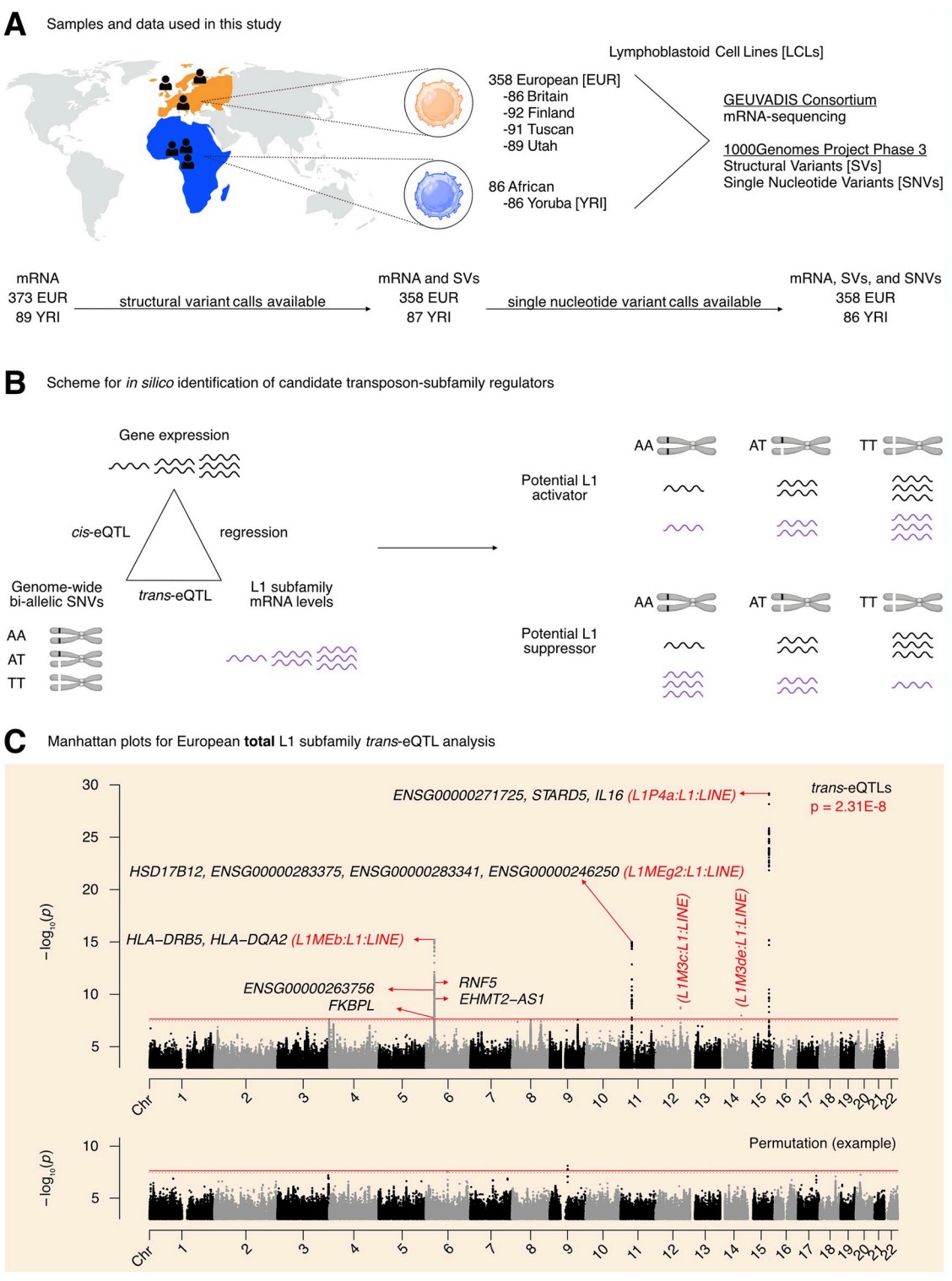

**Fig 1. Overview of the pipeline developed to scan for L1 transcriptional regulators *in silico*. (A)** An illustration of the samples and "omic" data used in this study. Of the 358 European individuals, 187 were female and 171 were male. Of the 86 African individuals, 49 were female and 37 were male. (Note that Utah subjects are of Northern European ancestry, and thus part of the European cohort for analytical purposes). **(B)** A schematic illustrating how genetic variants, gene expression, and TE expression can be integrated to identify highly correlated SNV-Gene-TE trios. **(C)** A Manhattan plot for the L1 subfamily *trans*-eQTL analysis in the European cohort. The genes that passed our three-part integration approach are listed next to the most significant *trans*-eQTL SNV they were

associated with in *cis*. The dashed line at p = 3.44E-8 corresponds to an average empirical FDR < 0.05, based on 20 random permutations. One such permutation is illustrated in the bottom panel. The solid line at p = 2.31E-8 corresponds to a Benjamini-Hochberg FDR < 0.05. The stricter of the two thresholds, p = 2.31E-8, was used to define significant *trans*-eQTLs. FDR: False Discovery Rate. Panels (**A**) and (**B**) were created with BioRender.com.

each L1 subfamily. We reasoned that there would have to be factors (i.e., miRNAs, proteins, non-coding RNAs) mediating at least a subset of the effects of SNVs on L1 subfamily RNA levels. Thus, to identify candidate genic mediators, we scanned for genome-wide gene *cis*-eQTLs, and then overlapped these with L1 *trans*-eQTLs. We note that Lappalainen et al. characterized gene *cis*-eQTLs in the original publication. However, we repeated the analysis using our pipeline and newer genome-wide SNV calls and annotations that were not available at the time, in order to obtain a homogenous and up-to-date set of *cis*- and *trans*-eQTL annotations that we could easily and reliably compare. As a final filter, we reasoned that for a subset of regulators, L1 subfamily RNA levels would respond to changes in the expression of those regulators. Consequently, we chose to quantify the association between L1 subfamily RNA levels and candidate gene expression by linear regression. Importantly, to avoid confounding eQTL associations with extraneous technical and biological factors, the expression data was corrected for the following: laboratory, population category, genetic population structure, biological sex, net L1 and Alu copy number called from the SV data, and EBV expression levels. We hypothesized that this three-part integration would result in combinations of significantly correlated SNVs, genes, and L1 subfamilies (Fig 1B).

The *trans*-eQTL analysis for RNA levels against every detected L1 subfamily led to the identification of 499 *trans*-eQTLs distributed across chromosomes 6, 11, 12, 14, and 15 that passed genome-wide significance (Fig 1C and Sheet A in S1 Table). The independent *cis*-eQTL analysis against all expressed genes led to the identification of 845,260 *cis*-eQTLs that passed genome-wide significance (S2A Fig and Sheet B in S1 Table). After integrating the identified *cis*- and *trans*-eQTLs and running linear regression, we identified 1,272 SNV-Gene-L1 trios that fulfilled our three-part integration approach (Sheet C in S1 Table). Among this pool of trios, we identified 7 unique protein-coding genes including (i) *IL16* and *STARD5* which were correlated with *L1P4a* levels (we note that some *L1P4a* fragments can be found in the introns of *IL16*), (ii) *HLA-DRB5*, *HLA-DQA2*, *RNF5*, and *FKBPL* which were correlated with *L1MEb* levels, and (iii) *HSD17B12* which was correlated with *L1MEg2* levels (Fig 1C). Although *EHMT2* did not pass our screening approach, it does overlap *EHMT2-AS1*, which did pass our screening thresholds. In contrast, we also identified "orphan" SNVs on chromosomes 12 and 14 which were associated with *L1M3c* and *L1M3de* levels in *trans* but to which we were unable to attribute a candidate gene. These SNVs resided in intronic regions within *NTN4* and *STON2*, respectively. We note that detection of these gene and TE associations is unlikely to be mechanistically related to variations in EBV expression, as expression profiles were corrected for such differences before downstream analyses (S2B Fig). We also note that several other unique non-coding genes, often overlapping the protein-coding genes listed, were also identified (Fig 1C). Finally, we recognize the possibility that unknown covariates, such as B cell activation state or EBV latency, may drive associations between SNVs and L1 RNA levels. To mitigate this possibility, we re-ran the *trans*-eQTL analysis with an additional 10 PEER factors [40], which can capture unknown drivers of variation. Though the absolute p-values were altered, this analysis also identified our previous top 3 loci on chromosomes 6, 11, and 15, as well as several additional peaks (S2C Fig and Sheet D in S1 Table). These results suggest that unknown covariates are unlikely to trivially explain the associations between SNVs at these loci and the corresponding L1 RNA levels. For simplicity of interpretation and reproducibility

reasons, during downstream experimental validation, we focused on protein-coding genes identified in the analysis without PEER factors (since the PEER package has not been maintained since R version 2.10).

Next, to define first and second tier candidate regulators, we clumped SNVs in linkage disequilibrium (LD) by L1 *trans*-eQTL p-value to identify the most strongly associated genetic variant in each genomic region (Figs 2A and S3A). LD-clumping identified the following index SNVs (*i.e.* the most strongly associated SNVs in a given region): rs11635336 on chromosome 15, rs9271894 on chromosome 6, rs1061810 on chromosome 11, rs112581165 on chromosome 12, and rs72691418 on chromosome 14 (Sheet E in S1 Table). Genes linked to these SNVs were considered first tier candidate regulators and included *IL16*, *STARD5*, *HLA-DRB5*, *HLA-DQA2*, and *HSD17B12* (Fig 2B and Sheet F in S1 Table). The remaining genes were linked to clumped, non-index SNVs and were consequently considered second tier candidates and included *RNF5*, *EHMT2-AS1*, and *FKBPL* (S3B Fig). For simplicity of interpretation, we considered only non-*HLA* genes during downstream experimental validation, since validation could be complicated by the highly polymorphic nature of *HLA* loci [41] and their involvement in multi-protein complexes. To gain insight into how index SNVs may influence *L1* expression, we compared the number of L1 fragments within 5 kb on either side of each index SNV and 1000 random SNVs (S3C Fig). We observed no significant difference between the two groups, suggesting that index SNVs are not in regions enriched for L1 insertions. Moreover, for the *IL16/STARD5* index SNV, no L1P4a copy was within the 10 kb window. These results suggest that differences in L1 expression are unlikely due to differences in proximal L1 copy number.

Finally, to computationally determine whether candidate genes may causally influence L1 subfamily RNA levels, we carried out mediation analysis on all SNV-gene-L1 trios (S4A Fig). Interestingly, 868 out of the 1,272 (68.2%) trios exhibited significant (FDR < 0.05) mediation effects (Sheet G in S1 Table). Among the 1st tier candidate regulators, significant, partial, and consistent mediation effects could be attributed to *STARD5*, *IL16*, *HSD17B12*, and *HLA-DRB5* (S4B Fig and Sheet G in S1 Table). To note, while significant mediation could be attributed to the index SNV for *STARD5*, significant mediation could only be attributed to clumped SNVs for *IL16* and *HSD17B12*. Given that *STARD5* and *IL16* share *cis*-eQTL SNVs, this suggests that *STARD5* may be the more potent mediator. Among the 2nd tier candidate regulators, significant, partial, and consistent mediation effects could be attributed to *RNF5*, *EHMT2-AS1*, and *FKBPL* (S4C Fig and Sheet G in S1 Table). These results suggest that candidate genes may mediate the effects between linked SNVs and L1 subfamilies.

## In silico scanning for L1 subfamily candidate regulators in an African population

We sought to assess the cross-ancestry regulatory properties of candidate genes by repeating our scan using the Yoruban samples as a smaller but independent replication cohort. Here, rather than conduct a genome-wide scan for *cis*- and *trans*- associated factors, we opted for a targeted approach focusing only on gene *cis*-eQTLs and L1 subfamily *trans*-eQTLs that were significant in the analysis with European samples (S5A Fig). The targeted *trans*-eQTL analysis led to the identification of 227 significant (FDR < 0.05) *trans*-eQTLs distributed across chromosomes 6 and 11 (Sheet A in S2 Table). The targeted *cis*-eQTL analysis led to the identification of 1,248 significant (FDR < 0.05) *cis*-eQTLs (Sheet B in S2 Table). After integrating the identified *cis*- and *trans*-eQTLs and running linear regression, we identified 393 SNV-Gene-L1 trios that fulfilled our three-part integration approach (Sheet C in S2 Table). Among this pool of trios, we identified 2 unique protein-coding genes—*HSD17B12 and HLA-DRB6*—as

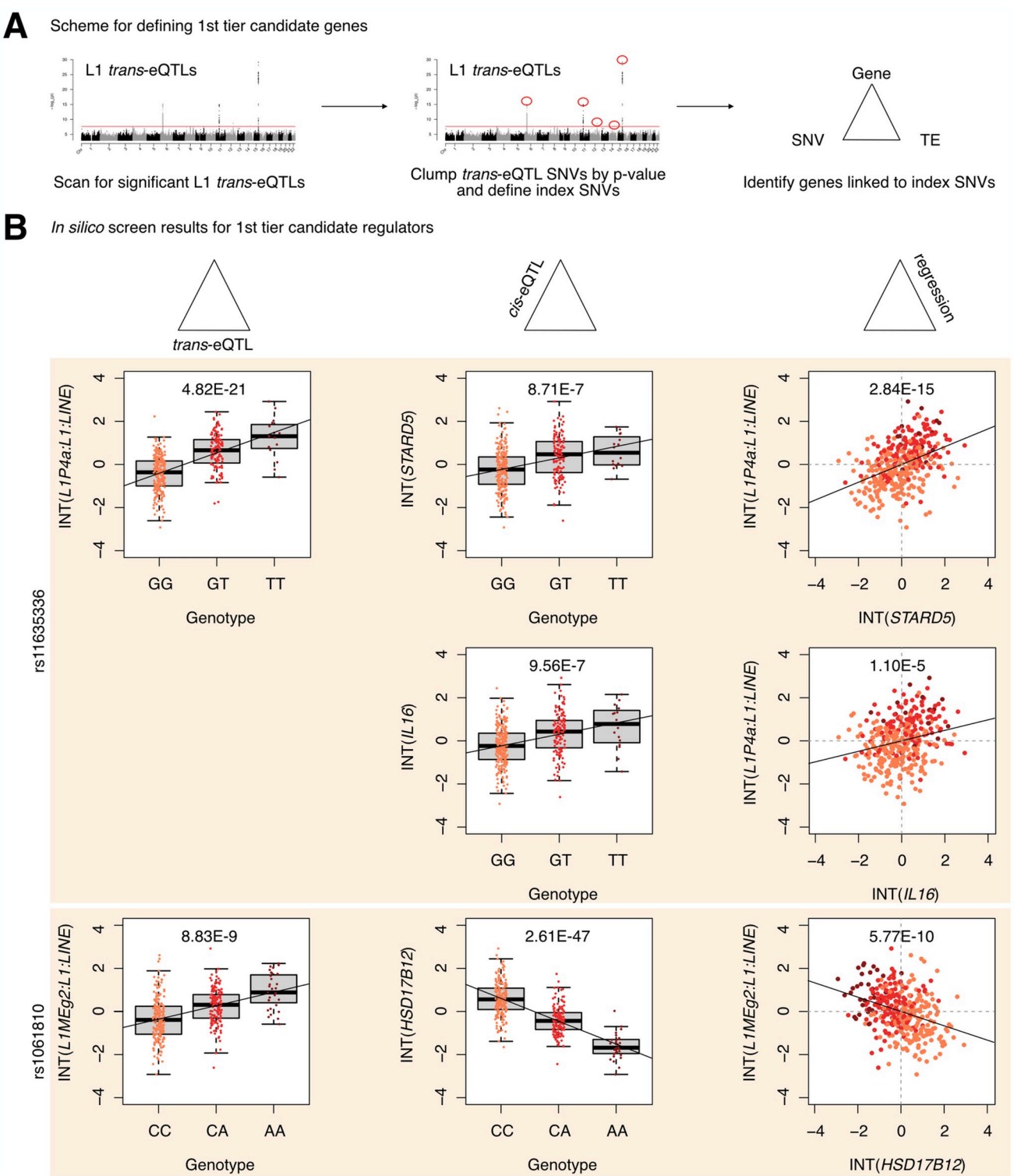

**Fig 2. Identification of 1st tier candidate L1 expression regulators in the European cohort.** (A) A schematic for how 1st tier candidate genes were defined. In short, these were genes in trios with index SNVs that were at the top of their respective peak. (B) The three-part integration results for three protein-coding genes—*STARD5*, *IL16*, *HSD17B12*—that we considered first tier candidates for functional, *in vitro* testing. In the left column are the *trans*-eQTLs, in the middle column are the *cis*-eQTLs, and in the right column are the linear regressions for gene expression against L1 subfamily expression. Expression values following an inverse normal transform (INT) are shown. The FDR for each analysis is listed at the top of each plot. FDR: False Discovery Rate.

well as several unique non-coding genes (Sheet C in S2 Table). Again, we clumped SNVs in linkage disequilibrium by L1 *trans*-eQTL p-value. LD-clumping identified the following index SNVs: rs2176598 on chromosome 11 and rs9271379 on chromosome 6 (Sheet D in S2 Table). Genes linked to these SNVs were considered first tier candidate regulators and included both *HSD17B12* and *HLA-DRB6* (S5B Fig and Sheet E in S2 Table). Finally, we carried out mediation analysis on all SNV-gene-L1 trios; however, no significant (FDR < 0.05) mediation was observed (Sheet F in S2 Table). These results implicate *HSD17B12* and the *HLA* loci as candidate, cross-ancestry regulators of L1 RNA levels.

To assess why some candidate genes did not replicate in the Yoruba cohort, we manually inspected *cis*- and *trans*-eQTL results for trios with those genes (S6A Fig). Interestingly, we identified rs9270493 and rs9272222 as significant (FDR < 0.05) *trans*-eQTLs for *L1MEb* RNA levels. However, those SNVs were not significant *cis*-eQTLs for *RNF5* and *FKBPL* expression, respectively. For trios involving *STARD5*, *IL16*, and *EHMT2-AS1*, neither the *cis*-eQTL nor the *trans*-eQTL were significant. We note that for most of these comparisons, although the two genotypes with the largest sample sizes were sufficient to establish a trending change in *cis* or *trans* RNA levels, this trend was often broken by the third genotype with spurious sample sizes. This suggests that replication in the Yoruba cohort may be limited by the small cohort sample size in the GEUVADIS project.

## Stratified in silico scanning for candidate regulators of intronic, intergenic, or exon-overlapping L1 subfamily RNA levels

One potential limitation with the approach undertaken thus far is the inability to distinguish different transposon RNA sources. For example, intergenic TEs may be transcribed using their own promoter or using a nearby gene's promoter. In contrast, while intronic or exon-overlapping TEs may be independently transcribed if they have retained their promoter, they may also appear expressed due to intron retention or due to exonization events. To have further granularity in deciphering L1 RNA level regulators with respect to the genomic provenance of their RNA sources, we next (i) carried out locus-specific quantification for each TE locus, (ii) stratified loci by whether they were intronic, nearby intergenic (within 5 kb of a gene), distal intergenic (>5 kb from a gene) or exon-overlapping, (iii) aggregated counts within each category at the subfamily level to compare with our unstratified TEtranscripts results, and (iv) repeated our eQTL scan using the four stratified TE RNA profiles (Sheets A–D in S3 Table and S7A–S7D Fig).

First, the eQTL scan using the intronic L1 profiles recapitulated the results of our initial scan for the *IL16/STARD5* and *HSD17B12* loci (S7A Fig and Sheet E in S3 Table). Interestingly, we recovered a dominant, new peak on chromosome 4 that was associated with *L1M3a* RNA levels but to which we could not attribute a protein-coding mediator. The index SNV for this locus, rs6819237, resides within an intron of *ZNF141*, which has been shown to bind L1PA elements [42]. Second, the eQTL scan using the nearby intergenic L1 profiles recapitulated the results of our initial scan for the *HLA* loci (S7B Fig and Sheet F in S3 Table). Third, for distal intergenic L1 expression, we identified a cluster of SNVs on chromosome 6 that were co-associated with *L1MC5* RNA levels and *ZSCAN26* expression (S7C Fig and Sheet G in S3 Table). Interestingly, these SNVs reside in a genomic region with many other *ZSCAN* genes, including *ZKSCAN4* which is hypothesized to regulate L1PA5/PA6 transcripts [43] and *ZSCAN9* which was shown to bind L1 by MapRRCon [44] analysis. Finally, we also identified several loci associated with exon-overlapping L1 RNA levels (S7D Fig and Sheet H in S3 Table).

Since intergenic L1s, as a potential source of independently transcribed L1 RNA, are of special interest, we repeated the mediation analysis for the *ZSCAN26*-associated SNV rs1361387

(S8A Fig). Alternating the genotype of rs1361387 was associated with an increase in *L1MC5* RNA levels and a decrease in *ZSCAN26* expression (S8B Fig). Mediation analysis revealed significant, but inconsistent mediation of *L1MC5* through *ZSCAN26* (S8C Fig and Sheet I in S3 Table). This may suggest that rs1361387 may exert both positive and negative control of *L1MC5* through uncharacterized mechanisms. Taken together, these results suggest that our approach can detect known L1 RNA regulators.

## TE families and known TE-associated pathways are differentially regulated across L1 trans-eQTL variants

Though our eQTL analysis identified genetic variants associated with the levels of specific, evolutionarily older L1 subfamilies, we reasoned that there may be more global but subtle differences in TE expression profiles among genotype groups, given that TE levels across subfamilies is highly correlated [21]. Thus, for each gene-associated index SNV identified in the European eQTL analysis, we carried out differential expression analysis for all expressed genes and TEs (Fig 3A). At the individual gene level, we detected few significant (FDR < 0.05) changes: 4 genes/TEs varied with rs11635336 genotype (*IL16*/*STARD5*), 4 genes/TEs varied with rs9271894 genotype (*HLA*), and 5 genes/TEs varied with rs1061810 genotype (*HSD17B12*) (Sheets A–C in S4 Table). Importantly, however, these genes/TEs overlapped the genes/TEs identified in the *cis*- and *trans*-eQTL analyses, providing an algorithm-independent link among candidate SNV-gene-TE trios.

In contrast to gene-level analyses, Gene Set Enrichment Analysis (GSEA) provides increased sensitivity to subtle, but consistent and widespread, transcriptomic changes at the level of gene sets (*e.g.* TE families, biological pathways, etc.) [45]. Specifically, GSEA was developed in response to (i) the lack of reproducibility of individual, significant gene changes across studies and (ii) the need to summarize biological changes in a functionally meaningful way through the use of gene sets containing biologically-related genes. For the TEs, we opted to aggregate TE subfamilies into gene sets corresponding to TE families on the basis that (i) broad changes in individual TE subfamilies may be hard to detect but changes across many subfamilies would be easier to detect, (ii) the expression of many different TE subfamilies was previously found to be highly correlated in an analysis of tumor-adjacent tissue [21], (iii) we were searching for factors influencing global TE RNA levels and not just specific TE loci, and (iv) GSEA has previously been applied to summarize TE changes [46–48]. We leveraged our differential expression analysis in combination with GSEA to identify repeat family and biological pathway gene sets impacted by SNV genotype in the GEUVADIS dataset (Sheets D–O in S4 Table and Fig 3A). Interestingly, changes in the genotype of rs11635336 (*IL16*/*STARD5*), rs9271894 (*HLA*), and rs1061810 (*HSD17B12*) were associated with an upregulation, upregulation, and downregulation, respectively, of multiple TE family gene sets (Fig 3B and Sheet P in S4 Table). Differentially regulated TE family gene sets included DNA transposons, such as the hAT-Charlie family, and long terminal repeat (LTR) transposons, such as the endogenous retrovirus-1 (ERV1) family (Fig 3B and Sheet P in S4 Table). Noteworthy, the L1 family gene set was the only TE gene set whose expression level was significantly altered across all three SNV analyses (Fig 3B and Sheet P in S4 Table). Consistent with their relative significance in the L1 *trans*-eQTL analysis, the L1 family gene set was most strongly upregulated by alternating the *IL16*/*STARD5* SNV (NES = 3.74, FDR = 6.43E-41), intermediately upregulated by alternating the *HLA* SNV (NES = 1.90, FDR = 7.19E-5), and least strongly changed by alternating the *HSD17B12* SNV (NES = -1.57, FDR = 2.11E-2) (Fig 3C). Among these changes, both older (L1M) and younger (L1PA) elements were differentially regulated across all three SNVs (Fig 3D and Sheet Q in S4 Table). Overall, we observed similar effects on TE family RNA levels

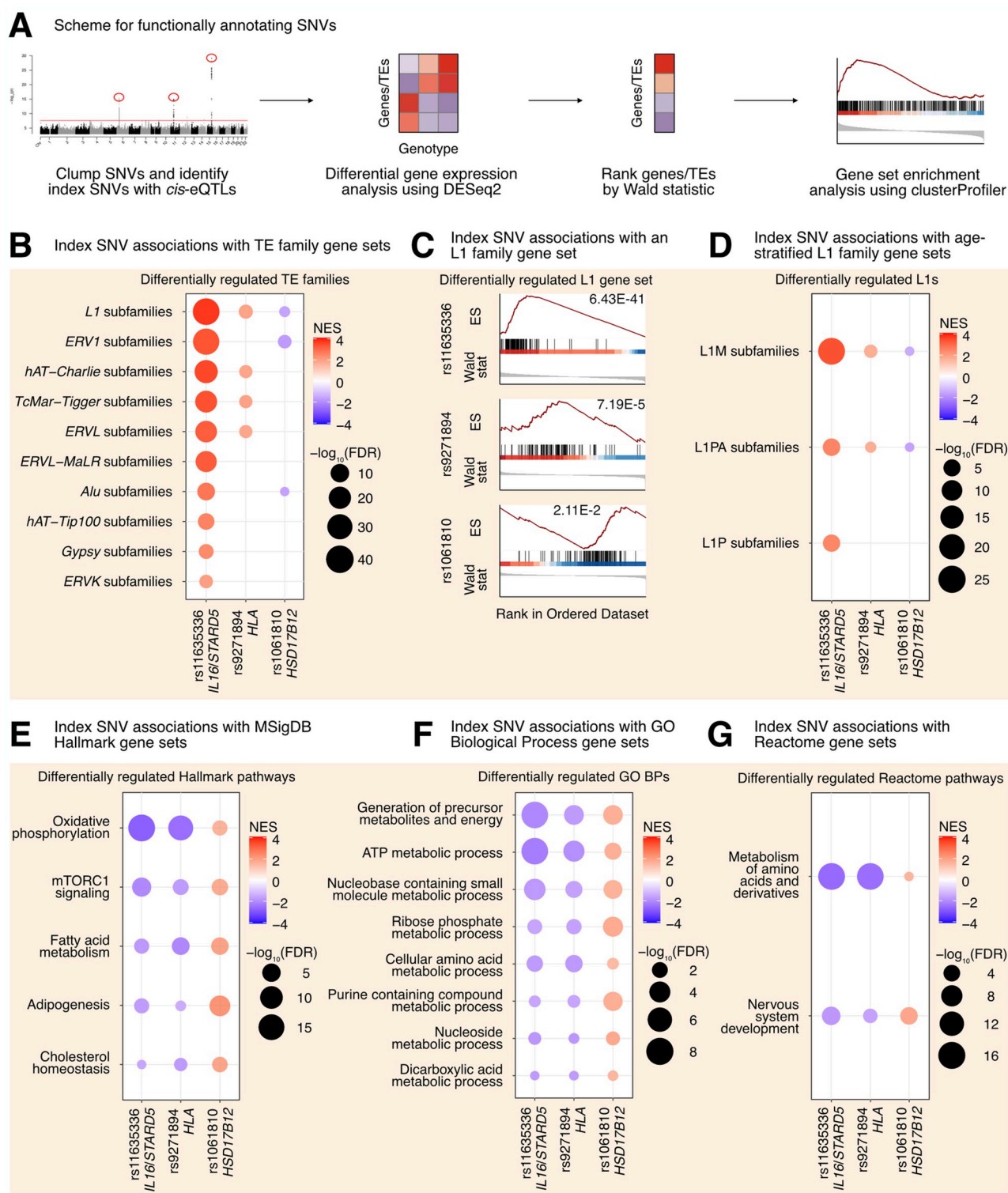

**Fig 3. L1 *trans*-eQTLs are associated with subtle, widespread differences in TE families and known TE-associated pathways.** (A) Scheme for functionally annotating gene-linked index SNVs by GSEA. (B) GSEA analysis for shared, significantly regulated TE family gene sets across genotypes for rs11635336 (*IL16/STARD5*), rs9271894 (*HLA*), and rs1061810 (*HSD17B12*). (C) GSEA plots for the L1 family gene set results summarized in (B). For these plots, the FDR value is listed. (D) GSEA analysis for shared, significantly regulated, evolutionary-age-stratified L1 gene sets across genotypes for rs11635336 (*IL16/STARD5*), rs9271894 (*HLA*), and rs1061810 (*HSD17B12*). L1M subfamilies are the oldest, L1P subfamilies are intermediate, and L1PA subfamilies are the youngest. GSEA analysis for top, shared, concomitantly regulated (E) MSigDB Hallmark pathway, (F) GO Biological Process,

and **(G)** Reactome pathway gene sets across genotypes for rs11635336 (*IL16/STARD5*), rs9271894 (*HLA*), and rs1061810 (*HSD17B12*). Shared gene sets were ranked by combining p-values from each individual SNV analysis using Fisher's method. In each bubble plot, the size of the dot represents the $-\log_{10}$(FDR) and the color reflects the normalized enrichment score. FDR: False Discovery Rate.

when, as an alternative and orthogonal approach, we applied a one-sample Wilcoxon test to determine whether TE family changes were significantly different than 0 across the three SNV DESeq2 analyses (S9A–S9C Fig). We briefly note here that rs9270493, a clumped SNV linked to *RNF5*, was also linked to upregulation of the L1 family gene set (Sheets R and S in S4 Table). These results suggest that TE subfamily *trans*-eQTLs are associated with subtle but global differences in TE RNA levels beyond a lone TE subfamily.

To determine the origin of these global TE RNA level differences, we repeated our differential expression analysis and GSEA using the genomic-region-stratified TE RNA profiles. Similar to the unstratified analysis, changes in the genotype of rs11635336 (*IL16/STARD5*), rs9271894 (*HLA*), and rs1061810 (*HSD17B12*) were associated with an upregulation, upregulation, and downregulation, respectively, of multiple TE family gene sets of varied genomic origin (Sheets T–V in S4 Table and S9D–S9F Fig). Across genotype for rs11635336 (*IL16/STARD5*), the most upregulated TEs were of intronic origin (S9D Fig). However, distal intergenic TE RNA levels, including L1 RNA levels, were also upregulated, suggesting that TE RNA level differences are not solely due to TE exonization or co-expression with genes. In contrast, most of the TE upregulation across genotypes for rs9271894 (*HLA*) was due to intronic TE RNA, though intergenic TEs near genes were also upregulated (S9E Fig). Finally, L1 RNA levels, and TE RNA levels more generally, were downregulated in the distal intergenic, nearby intergenic, and exonic categories for rs1061810 (*HSD17B12*) (S9F Fig). These results suggest that TE subfamily *trans*-eQTLs are associated with subtle but global differences in TE expression of varying genomic origin (intronic, exonic, or intergenic).

Next, we asked if other biological pathways were regulated concomitantly with TE gene sets in response to gene-linked index SNVs, reasoning that such pathways would act either upstream (as regulatory pathways) or downstream (as response pathways) of TE alterations. GSEA with the MSigDB Hallmark pathway gene sets [45,49] identified 5 gene sets fitting this criterion, including "oxidative phosphorylation", "mTORC1 signaling", "fatty acid metabolism", "adipogenesis", and "cholesterol homeostasis" (Fig 3E and Sheet W in S4 Table). Interestingly, several of these pathways or genes in these pathways have been implicated in TE regulation before. Rapamycin, which antagonizes mTORC1 function, has been shown to alter the expression of L1 and other repeats [32,50]. Specifically, Marasca et al. observed that naive CD4$^+$ T cells treated with rapamycin had higher L1 RNA levels compared to controls. In contrast, Wahl et al. observed a reduction of repeat RNA levels in the livers of mice treated with rapamycin. These differences highlight the need for more studies to define cell- and tissue-specific repeat RNA responses to treatments like rapamycin. Estrogens, which are involved in cholesterol and lipid metabolism, have also been found to drive changes in repeat expression, and the receptors for both estrogens and androgens are believed to bind repeat DNA [50,51]. Pharmacological inhibition of the mitochondrial respiratory chain and pharmacological reduction of endogenous cholesterol synthesis have also been shown to induce changes in L1 protein levels or repeat expression more broadly [52,53]. GSEA with the GO Biological Process gene sets (Fig 3F and Sheet X in S4 Table) and the Reactome gene sets (Fig 3G and Sheet Y in S4 Table) also identified several metabolism-related pathways including "ATP metabolic process", "Generation of precursor metabolites and energy", and "metabolism of amino acids and derivatives". These results add to the catalogue of pathways associated with differences in L1 expression.

In our eQTL analysis, we also identified two orphan index SNVs, rs112581165 and rs72691418, to which we could not attribute a protein-coding gene mediator. To determine whether these SNVs also regulate any transposon families or biological pathways, we repeated the differential expression analysis (with all expressed genes and TEs) (Sheets A and B in S5 Table) and the GSEA (Sheets C–J in S5 Table) with these SNVs (S10A Fig). At the individual gene level, we detected 3193 genes/TEs that varied significantly (FDR < 0.05) with rs112581165 genotype and 1229 genes/TEs that varied significantly with rs72691418 genotype (Sheets A and B in S5 Table). Similar to above, we next carried out GSEA to identify changes in functionally relevant gene sets. Like the gene-linked index SNVs, changes in the genotype of rs112581165 and rs72691418 were both associated with a downregulation and upregulation, respectively, of 10 TE families (S10B Fig and Sheet K in S5 Table). Noteworthy, the L1 family gene set was among the most strongly dysregulated TE family gene sets for both rs112581165 (NES = -4.32, FDR = 5.18E-89) and rs72691418 (NES = 4.01, FDR = 5.38E-79) (S10C Fig). Among L1 changes, older (L1M), intermediate (L1P), and younger (L1PA) elements were differentially regulated across both SNVs (S10D Fig and Sheet L in S5 Table).

Overall, we observed similar changes in TE RNA levels when we applied the alternative one-sample Wilcoxon test approach to determine whether TE family changes were significantly different than 0 across both SNV DESeq2 analyses (S11A and S11B Fig). After stratifying TE RNA levels by genomic origin, we observed that intronic TEs were strongly differentially regulated with genotype for both SNVs. However, differential regulation of intergenic TEs (both near and far from genes) and exonic TEs were also observed (Sheets M and N in S5 Table and S11C and S11D Fig). These results suggest that TE subfamily *trans*-eQTLs are associated with subtle differences in TE expression beyond the lone TE subfamily, even in the absence of a protein-coding gene *cis*-eQTL. Additionally, the data also suggests that TE RNA changes are not solely due to exonization or intron retention events.

Like before, we asked if other biological pathways were regulated concomitantly with TE gene sets in response to orphan index SNVs. The top 10 Hallmark pathway gene sets identified by GSEA included gene sets that were previously identified ("oxidative phosphorylation", "fatty acid metabolism", and "mTORC1 signaling"), as well as several new pathways (S10E Fig and Sheet O in S5 Table). Among the new pathways, "DNA repair" [20] and the "P53 pathway" [37,54] have also been linked to L1 control, and proteins in the "Myc targets v1" gene set interact with L1 ORF1p [36]. GSEA with the GO Biological Process gene sets (S10F Fig and Sheet P in S5 Table) and the Reactome gene sets (S10G Fig and Sheet Q in S5 Table) identified several metabolism-related pathways and several translation-related pathways, such as "cytoplasmic translation", "eukaryotic translation initiation", and "eukaryotic translation elongation". Importantly, proteins involved in various aspects of proteostasis have been shown to be enriched among L1 ORF1p-interacting proteins [36]. Again, these results add to the catalogue of pathways associated with differences in TE expression, even in the absence of a candidate *cis* mediator.

Finally, we carried out our DESeq2 and GSEA analysis against the lone index SNV, rs1361387, associated with distal intergenic L1 RNA levels (S12A Fig and Sheets R–U in S5 Table). For this SNV, we did not detect significant changes in any TE family gene set. However, GSEA with all three pathway gene sets revealed a strong suppression of immune related processes, including "interferon gamma response", "interferon alpha response", "response to virus", and "interferon alpha/beta signaling" (S12B–S12D Fig). These observations are consistent with the role of L1 as a stimulator and target of the interferon pathway [15,36], as well as the notion that transposons can mimic viruses and stimulate immune responses from their hosts [55].

## Modulation of top candidate gene activity in a lymphoblastoid cell line induces small but widespread TE RNA level changes

We decided to validate the L1 regulatory properties of top candidate genes associated with L1 *trans*-eQTLs. For experimental purposes, we selected the GM12878 lymphoblastoid cell line, because (i) it is of the same cell type as the transcriptomic data used here for our eQTL analysis, and (ii) its epigenomic landscape and culture conditions have been well well-characterized as part of the ENCODE project [56,57]. For validation purposes, we selected *IL16*, *STARD5*, *HSD17B12*, and *RNF5* out of the 7 protein-coding gene candidates. We chose these genes for validation because the first 3 are associated with top *trans*-eQTL SNVs and the fourth one had very strong predicted mediation effects. To note, although GM12878 was part of the 1000Genomes Project, it was not included in the GEUVADIS dataset. However, based on its genotype, we can predict the relative expression of candidate regulators (S13A and S13B Fig), which suggest that GM12878 may be most sensitive to modulations in *IL16* and *STARD5* expression, given their relatively low, predicted endogenous expression. Interestingly, examination of the ENCODE epigenomic data in GM12878 cells [56] demonstrated that the region near the *IL16*/*STARD5*-linked index SNV (rs11635336) was marked with H3K4Me1 and H3K27Ac, regulatory signatures of enhancers (S13C Fig). Similarly, the region near the *HLA*-linked index SNV (rs9271894) was marked with H3K4Me1, marked with H3K27Ac, and accessible by DNase, suggesting regulatory properties of the region as an active enhancer (S13C Fig). These results further highlight the regulatory potential of the *IL16*-, *STARD5*-, and *HLA*-linked SNVs.

First, we tested the transcriptomic impact of overexpressing our top candidates in GM12878 LCLs. Cells were electroporated with overexpression plasmids (or corresponding empty vector), and RNA was isolated after 48 hours (Figs 4A and S14A). Differential expression analysis comparing control and overexpression samples confirmed the overexpression of candidate genes (S14B Fig and Sheets A–D in S6 Table). We note that no significant differences in EBV expression were identified in any of the four conditions (all FDR > 0.05; S14C Fig). Intriguingly, we observed that *IL16* was significantly upregulated following *STARD5* overexpression (S14D Fig and Sheet B in S6 Table), although the inverse was not observed (Sheet A in S6 Table), suggesting that *IL16* may act downstream of *STARD5*. We note here that, consistent with the use of a high expression vector, the *IL16* upregulation elicited by *STARD5* overexpression ($\log_2$ fold change = 0.45) was weaker than the upregulation from the *IL16* overexpression ($\log_2$ fold change = 1.89) (Sheets A and B in S6 Table).

To further assess the biological relevance of each overexpression, we carried out GSEA using the GO Biological Process, Reactome pathway, and Hallmark pathway gene sets (Sheets E–P in S6 Table). Importantly, GSEA using GO Biological Process and Reactome pathway gene sets highlighted differences that were consistent with the known biology of our candidate genes. Firstly, *IL16* is involved in regulating T-cell activation, B-cell differentiation, and functions as a chemoattractant [58–63]. Moreover, it modulates macrophage polarization by regulating *IL-10* expression [64]. *IL16* overexpressing cells showed upregulation for "phagocytosis recognition" and "positive chemotaxis", downregulation for "negative regulation of cell differentiation", and downregulation for "Interleukin-10 signaling" (Fig 4B and 4C). Secondly, *STARD5* encodes a cholesterol transporter and is upregulated in response to endoplasmic reticulum (ER) stress [65–67]. *STARD5* overexpressing cells showed downregulation of various cholesterol-related gene sets such as "sterol biosynthetic process", "sterol metabolic process", and "regulation of cholesterol biosynthesis by SREBP (SREBF)" (Fig 4D and 4E). Thirdly, *HSD17B12* encodes a steroid dehydrogenase involved in converting estrone into estradiol and is essential for proper lipid homeostasis [68–70]. *HSD17B12* overexpressing cells showed downregulation of cholesterol-related gene sets, including "sterol biosynthetic

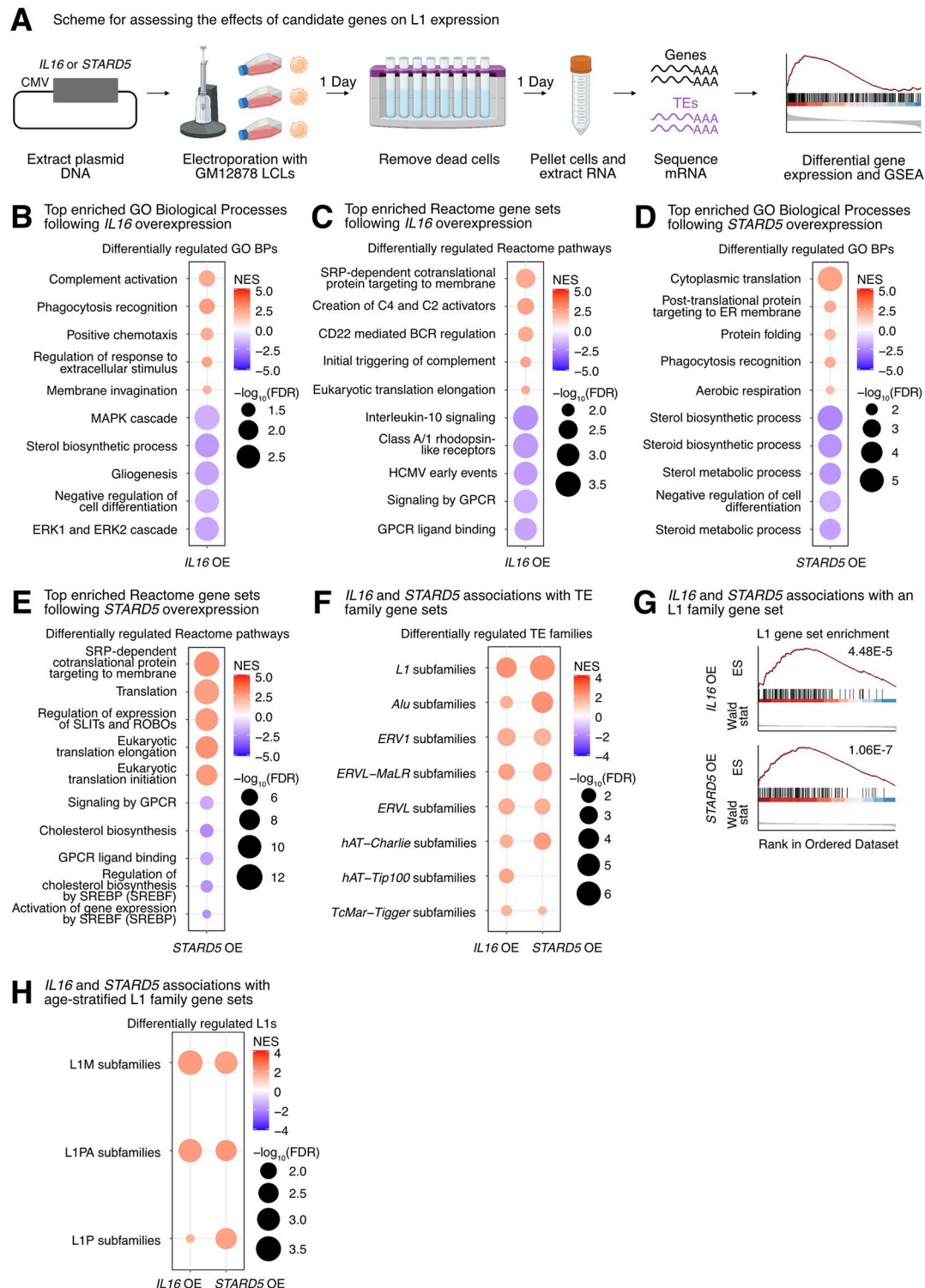

**Fig 4. Impact of *IL16* and *STARD5* overexpression on LCL gene and TE expression landscapes.** *IL16* and *STARD5* overexpression induce changes consistent with their known biology, as well as subtle but widespread upregulation of TE families. (**A**) Scheme for experimentally validating the roles of *IL16* and *STARD5* in L1 regulation. GSEA analysis for top, differentially regulated (**B**) GO Biological Process and (**C**) Reactome pathway gene sets following *IL16* overexpression. GSEA analysis for top, differentially regulated (**D**) GO Biological Process and (**E**) Reactome pathway gene sets following *STARD5* overexpression. (**F**) GSEA analysis for shared, significantly regulated TE family gene sets following *IL16* and *STARD5*

overexpression. **(G)** GSEA plots for the L1 family gene set results summarized in **(F)**. For these plots, the FDR value is listed. **(H)** GSEA analysis for shared, significantly regulated, evolutionary-age-stratified L1 gene sets across *IL16* and *STARD5* overexpression. L1M subfamilies are the oldest, L1P subfamilies are intermediate, and L1PA subfamilies are the youngest. In each bubble plot, the size of the dot represents the -log$_{10}$(FDR) and the color reflects the normalized enrichment score. FDR: False Discovery Rate. Panel **(A)** was created with BioRender.com.

process" and "sterol metabolic process" (S14E Fig). Finally, *RNF5* encodes an ER and mitochondrial-bound E3 ubiquitin-protein ligase that ubiquitin-tags proteins for degradation [71–74]. *RNF5* overexpressing cells demonstrated alterations in gene sets involved in proteostasis and ER biology, including upregulation of "ERAD pathway" and "response to endoplasmic reticulum stress" (S14F Fig). These results suggest that our approach leads to biological changes consistent with the known biological impact of the genes being overexpressed.

Next, we sought to determine whether modulation of candidate genes had any impact on TE RNA levels in general, and L1 in particular. Although there were no significant changes for individual TE subfamilies following *IL16* and *STARD5* overexpression (Sheets A and B in S6 Table), we identified subtle but widespread upregulation of various TE families across both conditions by GSEA (Fig 4F and Sheets Q and R in S6 Table). Interestingly, 7 families, including L1, ERV1, ERVL-MaLR, Alu, ERVL, TcMar-Tigger, and hAT-Charlie families, were commonly upregulated under both conditions (Fig 4F). In contrast, cells overexpressing *HSD17B12* or *RNF5* did not drive widespread changes in L1 family expression as assessed by GSEA (Sheets S and T in S6 Table), suggesting specificity of the *IL16/STARD5*-L1 relationships. Noteworthy, the L1 family gene set was more significantly upregulated following *STARD5* overexpression (NES = 2.27, FDR = 1.06E-7) compared to *IL16* overexpression (NES = 2.27, FDR = 4.48E-5) (Fig 4G and Sheets Q and R in S6 Table). Since *IL16* is upregulated in response to *STARD5* overexpression, this suggests that *STARD5* may synergize with *IL16* for the regulation of L1 RNA levels.

Overall, we observed similar changes in TE RNA levels when we applied the alternative one-sample Wilcoxon test approach to determine whether TE family changes were significantly different than 0 across overexpression conditions (S15A and S15B Fig). Among L1 changes, older (L1M), intermediate (L1P), and younger (L1PA) elements were differentially regulated across both overexpression conditions (Fig 4H and Sheet U in S6 Table). To gain insight into the mechanism of *IL16/STARD5*-mediated TE mis-regulation, we again stratified the TE expression profiles by genomic origin (Sheets V and W in S6 Table). Though intronic TEs of various families were strongly upregulated following *IL16* and *STARD5* overexpression, distal intergenic L1 RNA levels were also upregulated in both conditions (S15C and S15D Fig). These results further suggest that *IL16* and *STARD5* influence the repetitive RNA pools, including elements that are unlikely to be transcribed by neighboring genes.

Then, we decided to further characterize the impact of IL16 activity on TEs, since (i) its overexpression led to a global upregulation of TE transcription, and (ii) it was itself upregulated in response to *STARD5* overexpression, which also led to increased TE expression. Thus, since IL16 is a soluble cytokine, we independently assessed its regulatory properties by exposing GM12878 cells to recombinant human IL16 peptide [rhIL16] for 24 hours (Figs 5A and S16A). Differential gene expression analysis (Sheet A in S7 Table) and comparison with the *IL16* overexpression results demonstrated that differentially expressed genes were weakly but significantly correlated (S16B Fig). As with the overexpression conditions, no significant differences in EBV expression were identified (FDR > 0.05; S16C Fig). Additionally, we carried out GSEA using the GO Biological Process, Reactome pathway, Hallmark pathway, and TE family gene sets (Sheets B–E in S7 Table) and compared those results with the GSEA from the

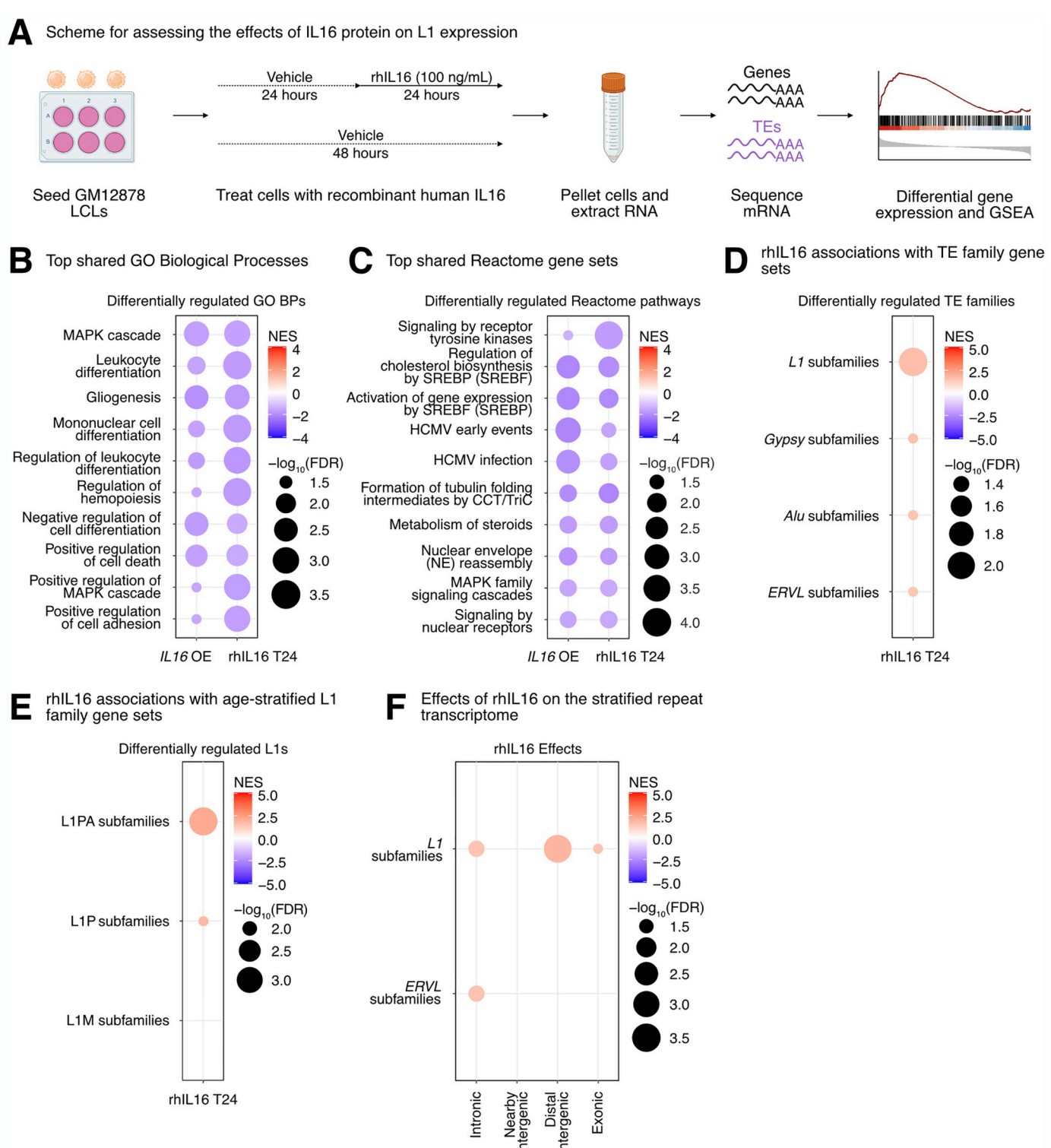

**Fig 5. rhIL16 treatment is sufficient to transiently upregulate an L1 family gene set.** (**A**) Scheme for experimentally validating the role of rhIL16 in L1 regulation. GSEA analysis for top, shared, concomitantly regulated (**B**) GO Biological Process and (**C**) Reactome pathway gene sets following *IL16* overexpression and rhIL16 exposure for 24 hours. Shared gene sets were ranked by combining p-values from each individual treatment analysis using Fisher's method. (**D**) GSEA analysis for top, differentially regulated TE family gene sets following rhIL16 exposure for 24 hours. (**E**) GSEA analysis for significantly regulated evolutionary-age-stratified L1 gene sets following rhIL16 exposure for 24 hours. L1M subfamilies are the oldest, L1P subfamilies are intermediate, and L1PA subfamilies are the youngest. (**F**) GSEA analysis for top, differentially regulated TE family gene sets in different genomic locations following rhIL16 exposure for 24 hours. In each

bubble plot, the size of the dot represents the -log$_{10}$(FDR) and the color reflects the normalized enrichment score. FDR: False Discovery Rate. Panel (**A**) was created with BioRender.com.

*IL16* overexpression (Sheets F–H in S7 Table). Consistent with the known biology of *IL16*, GSEA highlighted a downregulation of many immune cell-related gene sets such as "leukocyte differentiation" and "mononuclear cell differentiation" (Fig 5B and 5C and Sheets F–H in S7 Table). Similar to our overexpression results, exposure of GM12878 to rhIL16 for 24 hours led to the upregulation of an L1 family gene set by GSEA, although the effect was less pronounced than with the overexpression (Fig 5D). Again, we observed similar changes in TE RNA levels when we applied the one-sample Wilcoxon test alternative approach to determine whether TE family changes were significantly different than 0 following rhIL16 treatment for 24 hours (S16D Fig). Similar to the overexpression, intermediate (L1P) and younger (L1PA) age L1 elements were upregulated following rhIL16 treatment for 24 hours (Fig 5E and Sheet I in S7 Table). After stratifying the TE RNA profiles by genomic origin and running GSEA, we again observed an upregulation of intronic and distal intergenic L1 RNA, as in the overexpression (Fig 5F and Sheet J in S7 Table). Even though treatment of GM12878 with rhIL16 for 48 hours exhibited known features of IL16 biology (S16B, S16E and S16F Fig and Sheets K–S in S7 Table), the L1 upregulation was no longer detectable, though other TEs remained upregulated (S16G Fig and Sheet S in S7 Table). These results further support the notion that *IL16* acts as a modulator of L1 RNA levels, including for both intronic and distal intergenic copies. Importantly, our initial eQTL analysis identified a strong association between intronic L1 RNA levels and variants near *IL16/STARD5*. However, the *IL16* overexpression, *STARD5* overexpression, and rhIL16 peptide treatment all independently influenced *intergenic* L1 expression (in addition to intronic L1 expression). Since both the overexpression constructs and the peptide lack the *L1P4a* intronic sequence, our data suggests that these effects can be attributed to the protein-coding sequences, specifically. These results provide experimental evidence that *IL16* and *STARD5* may broadly impact the levels of both intronic and non-intronic L1 RNA levels.

Finally, we sought to define the biological pathways regulated concomitantly with the L1 family gene set under all experimental conditions where it was upregulated (i.e., *IL16* overexpression, *STARD5* overexpression, and 24 hours of rhIL16 exposure) (Fig 6A and 6B and Sheet A in S8 Table). Again, we reasoned that such pathways would act either upstream (as regulatory pathways) or downstream (as response pathways) of TE alterations. GSEA with the Hallmark pathway gene sets identified 7 gene sets fitting this criterion, including "TNFα signaling via NF-KB", "IL2 STAT5 signaling", "inflammatory response", "mTORC1 signaling", "estrogen response early", "apoptosis", and "UV response up" (Fig 6C and Sheet B in S8 Table). GSEA with the GO Biological Process gene sets (Fig 6D and Sheet C S8 Table) and the Reactome pathway gene sets (Fig 6E and Sheet D in S8 Table) also identified MAPK signaling, virus-related pathways like "HCMV early events", pathways involved in cell differentiation, and pathways involved in cholesterol and steroid metabolism like "signaling by nuclear receptors". These results further cement the catalogue of pathways associated with differences in TE RNA levels.

## L1 trans-eQTLs are co-associated with aging traits in GWAS databases

Although TE de-repression has been observed broadly with aging and age-related disease [5,75], whether this de-repression acts as a causal driver, or a downstream consequence, of aging phenotypes remains unknown. We reasoned that if increased TE expression at least partially drives aging phenotypes, L1 *trans*-eQTLs should be enriched for associations to aging

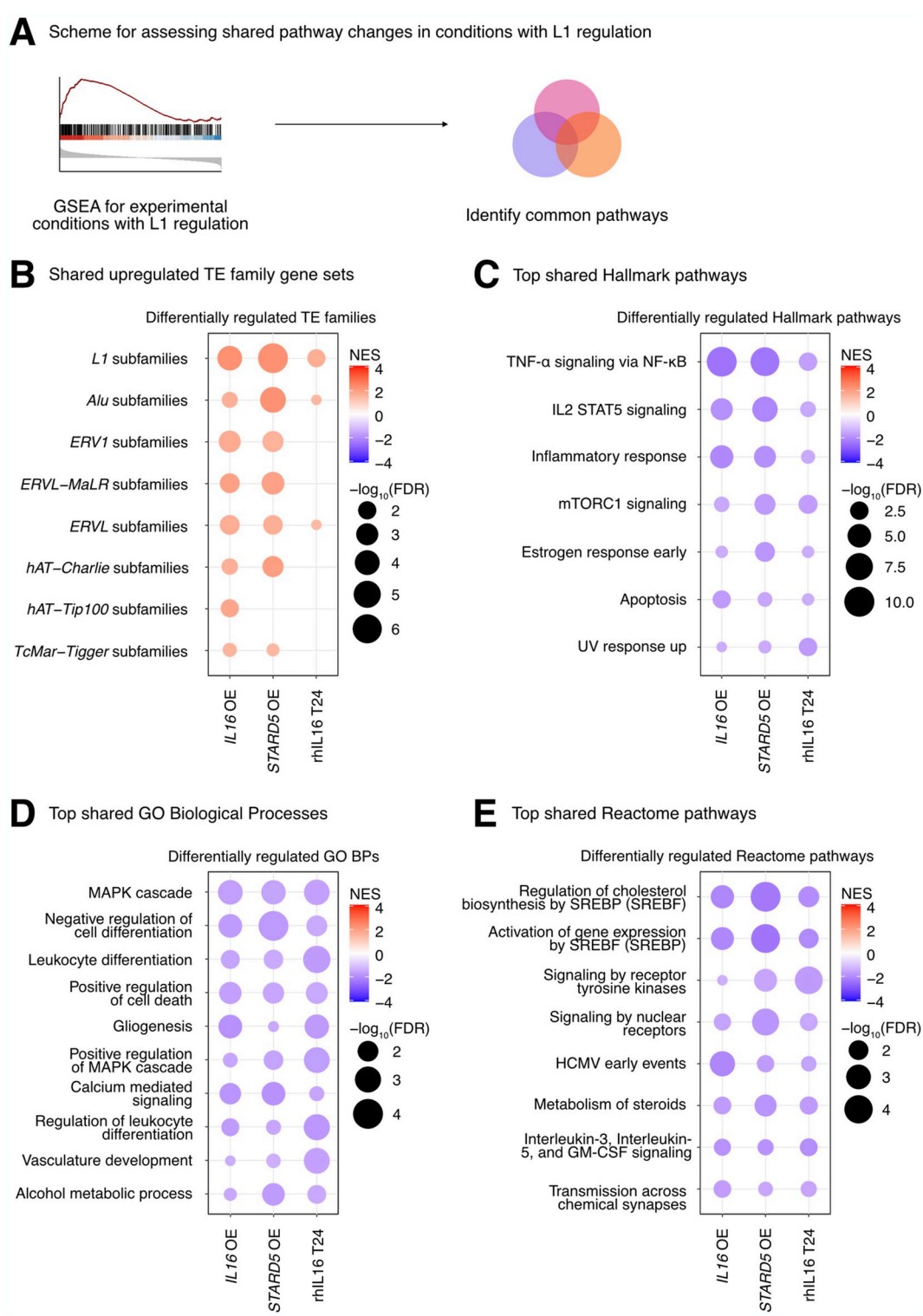

**Fig 6. Consistent cellular responses to *IL16* overexpression, *STARD5* overexpression, and rhIL16 exposure for 24 hours.**
*IL16* overexpression, *STARD5* overexpression, and rhIL16 exposure for 24 hours are associated with subtle but widespread differences in TE families and known TE-associated pathways. **(A)** Scheme for assessing concordantly regulated TE family and pathway gene sets across conditions where an L1 gene set is upregulated. GSEA analysis for top, shared, concomitantly regulated **(B)** TE family, **(C)** MSigDB Hallmark pathway, **(D)** GO Biological Process, and **(E)** Reactome pathway gene sets following *IL16* overexpression, *STARD5* overexpression, and rhIL16 exposure for 24 hours. Shared gene sets were ranked by

combining p-values from each individual treatment analysis using Fisher's method. In each bubble plot, the size of the dot represents the -log$_{10}$(FDR) and the color reflects the normalized enrichment score. FDR: False Discovery Rate.

traits in genome-wide association studies [GWAS] or phenome-wide association studies [PheWAS].

To test our hypothesis, we queried the Open Targets Genetics platform with our initial 499 *trans*-eQTL SNVs, mapped traits to standardized MeSH IDs, and then manually curated MeSH IDs related to aging-related traits (Fig 7A). Consistent with our hypothesis, a large proportion of L1 *trans*-eQTL SNVs (222/499 or 44.5%) were either (i) associated with an aging MeSH trait by PheWAS or (ii) LD-linked to a lead variant associated with an aging MeSH trait (Fig 7B). Moreover, among the 222 SNVs with significant aging-trait associations, we observed frequent mapping to more than a single age-related trait by PheWAS, with many SNVs associated with 10–25 traits (Fig 7C and Sheet A in S9 Table). Additionally, many of the 222 SNVs mapped to 1–5 aging traits through a proxy lead variant (Fig 7D and Sheet A in S9 Table). Among the most frequently associated or linked traits, we identified type 2 diabetes mellitus, hyperparathyroidism, thyroid diseases, coronary artery disease, hypothyroidism, and psoriasis, among many others (Fig 7E and Sheet B in S9 Table).

As a parallel approach, we queried the Open Targets Genetics platform with our L1 *trans*-eQTL SNVs, as well as 500 combinations of random SNVs sampled from all SNVs used in the eQTL analyses. We then leveraged broader phenotype categories annotated by the platform, including 14 disease categories that we considered aging-related, to determine whether L1 eQTL associations were enriched for any disease categories (S17A Fig). L1 eQTL associations were significantly enriched (FDR < 0.05 and ES > 1) for 13 out of 14 disease categories, including cell proliferation disorders, immune system diseases, and musculoskeletal diseases (S17B–S17N Fig). The cardiovascular diseases category was the only disease category for which we did not observe a significant enrichment (S17O Fig). The enrichment for cell proliferation disorders is consistent with the associations of L1 activity with cellular senescence [12,15] and cancer [76,77]. The enrichment for immune system diseases is consistent with the role of L1 as a stimulator of the interferon pathway, inflammation, and senescence [15], as well as the more general notion that transposons can mimic viruses and stimulate immune responses from their hosts [55]. The enrichment for musculoskeletal diseases is consistent with an increase in L1 expression and copy number with age in muscle tissue from aging mice [11]. These results reinforce the notion that L1 activity is strongly and non-randomly associated with an assortment of age-related diseases.

Intriguingly, a large fraction of co-associated SNVs were on chromosome 6 near the HLA locus, which has previously been shown to be a hotspot of age-related disease traits [78]. Despite its association to our strongest L1 trans-eQTL SNV, little is known about the regulation and impact of IL16 during aging. One study, however, found that *IL16* expression increases with age in ovarian tissue, and the frequency of *IL16* expressing cells is significantly higher in ovarian tissue from women at early and late menopause, compared to premenopausal women [79]. Given these findings, and since L1 expression levels and copy number have been found to increase with age [reviewed in [5]], we asked whether circulating IL16 levels may also change with age, using C57BL/6JNia mice as a model (Fig 7F and Sheet C in S9 Table). Consistent with the notion that increased IL16 levels may, at least partially, drive age-related TE de-repression, we observed a significant increase in circulating IL16 levels in female mice with age, and a trending increase with age in male mice (although the levels showed more animal-to-animal variability). By meta-analysis, circulating IL16 levels changed

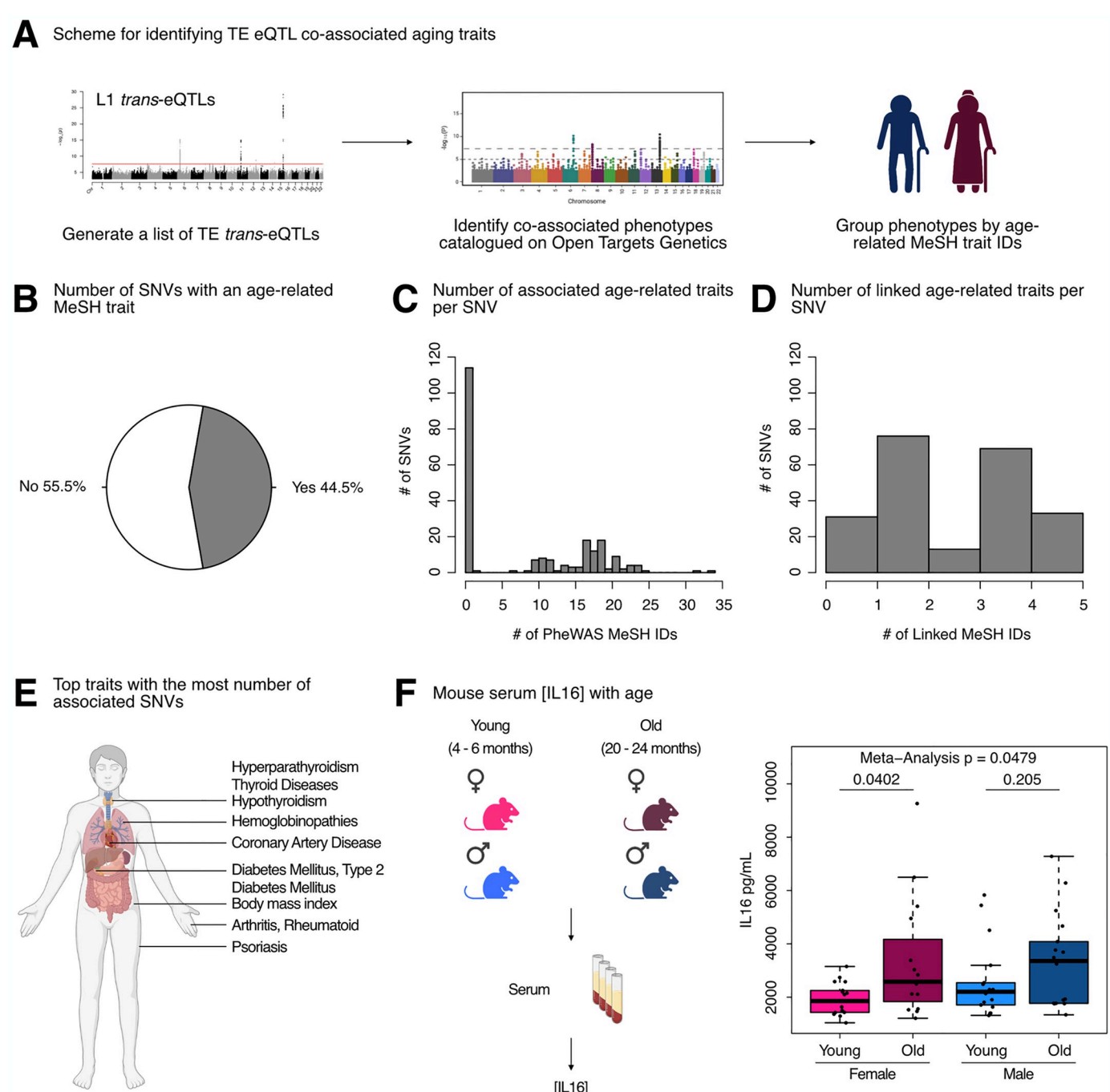

**Fig 7. L1 *trans*-eQTLs are co-associated with aging traits in GWAS databases. (A)** Scheme for obtaining *trans*-eQTL SNV-associated aging phenotypes from the Open Targets Genetics platform. **(B)** A pie chart representing the number of SNVs (222/499) associated with an aging-related MeSH trait, either by PheWAS or indirectly linked to the phenotype through a proxy lead SNP in LD with the SNV. **(C)** Histogram depicting the distribution of number of aging MeSH traits associated with the 222/499 SNVs by PheWAS. **(D)** Histogram depicting the distribution of number of aging MeSH traits linked with the 222/499 SNVs through a proxy lead SNP in LD with the SNVs. **(E)** A diagram highlighting the organ targets of the top 10 most frequently associated aging traits. **(F)** The concentrations of circulating IL16 in aging mice of both sexes was assessed by ELISA. Each dot represents an independent animal, with n = 15–17 per group. Significance across age in each sex was assessed using a Wilcoxon test. The p-values from each sex (females in pink and males in blue) were combined by meta-analysis using Fisher's method. Any p-value < 0.05 was considered significant. Panels **(A)**, **(E)**, and **(F)** were created with BioRender.com.

significantly with age across sexes (Fig 7F). These results further support the hypothesis that *IL16* is involved in L1 biology and may modulate L1 age-related changes. In sum, our results provide one of the first pieces of evidence of a causal link between L1 RNA levels and age-related decline.

## Discussion

### A new approach to identify regulators of TE expression

In this work, we developed a pipeline to computationally identify candidate L1 RNA level regulators by eQTL analysis. We provide experimental evidence for the involvement of top candidates in regulating L1 RNA levels, demonstrating as a proof-of-principle that this approach can be broadly used on other large "omic"-characterized cohorts with human (i.e. GTEx [80,81] or HipSci [82]) or mouse (i.e. DO mice [83]) subjects to identify other regulators of L1 activity. These datasets, combined with our approach, could be utilized to rigorously characterize conserved or group-specific TE regulatory mechanisms on multiple layers, such as across TE families (like Alu or ERVs), across cell or tissue types, across ancestry groups, and across species. This approach, which leverages existing datasets to perform *in silico* screening, could be a powerful method to expand our knowledge of TE regulation in non-diseased cells and tissues.

Though our initial scan identified genetic variants associated with expression differences in specific L1 subfamilies, secondary analyses by GSEA suggest that genetic variants are associated with subtle but global differences in the expression of many TE families of varying genomic context, including intronic, intergenic, and exonic TE RNA levels. Our pipeline identified candidate genes, including *HSD17B12* and *HLA* genes, that likely play a conserved role in L1 regulation across human populations of different ancestries. Though some top candidates from the European cohort scan, such as *IL16*, *STARD5*, and *RNF5*, were not significant in the African cohort analysis, it is likely that some of these genes would appear in cross-ancestry scans with larger samples sizes. To note, none of our top candidates were associated with L1 polymorphisms in the landmark TE-eQTL study that first used this data to study TE biology [27], suggesting that our findings are likely mechanistically independent. Importantly, we recognize that in projects of the same magnitude as the GEUVADIS consortium's project, it is nearly impossible to avoid introducing batch effects and to measure all possible covariates. Thus, one caveat with the eQTL approach is that unknown factors, such as B cell activation state or EBV latency, may confound SNV-L1 expression relationships. We attempted to mitigate this by employing PEER factors to account for unknown drivers of variation. Importantly, this analysis recovered the same top 3 loci as the analysis without PEER factors. Thus, the SNV-L1 associations are likely real and not due to external, unknown trivial factors.

After repeating our computational scan using the TE profiles stratified by genomic context, we made a number of additional insights, such as (i) the association of *IL16* and *STARD5* with intronic L1 RNA levels, (ii) the association of *HLA* with nearby intergenic L1 RNA levels, (iii) the identification of an additional candidate regulator, *ZSCAN26*, which may influence distal intergenic L1 RNA levels, and (iv) a number of SNVs associated with exon-overlapping L1 RNA levels. Moreover, the secondary GSEA analyses suggest that the SNVs tested exert broad effects on TE family RNA levels, regardless of genomic context.

As an important aspect of this study, we experimentally validated our top candidate genes. We detected subtle but global differences in L1 family RNA levels following *IL16* overexpression, *STARD5* overexpression, and rhIL16 treatment for 24 hours, further suggesting that our top candidates have regulatory potential. Although *IL16/STARD5* were mainly associated to intronic L1 levels in our trans-eQTL analysis, these treatments affected both intronic and distal

intergenic L1 RNA levels, suggesting that these differences are not solely due to intron retention or co-expression with neighboring genes. Surprisingly, our treatments exerted effects on other TE families as well, suggesting broad alterations that promote TE RNA differences. Thus, *IL16* and *STARD5* are likely to be *bona fide* regulators that can be prioritized for follow-up study.

Much of the work on L1 regulation relies on overexpressing full-length L1 elements in cell lines that can tolerate these manipulations [20,31]. However, there are some approaches aimed at characterizing transcription factors that bind endogenous L1 promoters [44,84], in addition to those that implement gene network approaches to find potential regulators that may act through alternative mechanisms [21]. These, in our view, complement plasmid-based approaches through a more physiological study of L1 regulators. We think that our approach, which relies on endogenous TE profiles, adds to this category of tools.

## New candidate L1 regulators are involved in viral response

As another, theoretical line of evidence for the potential involvement of our top candidate genes in L1 regulation, we highlight known interactions between tested candidate genes and viral infections, which may be relevant under conditions where transposons are recognized as viral mimics [55]. Indeed, *IL16* has been extensively studied for its ability to inhibit human immunodeficiency virus (HIV) replication, partly by suppressing mRNA expression [85–87]. Additionally, but in contrast to its HIV-suppressive properties, *IL16* can enhance the replication of influenza A virus (IAV) and facilitate its infection of hosts, potentially through its repression of type I interferon beta and interferon-stimulated genes [88]. *IL16* can also contribute to the establishment of lifelong gamma herpesvirus infection [89]. *STARD5* is another candidate implicated in the influenza virus replication cycle [90]. *HSD17B12* promotes the replication of hepatitis C virus via the very-long-chain fatty acid (VLCFA) synthesis pathway and the production of lipid droplets important for virus assembly [91,92]. Additionally, HSD17B12 has been found interacting with the coronavirus disease 2019 (COVID-19) protein nonstructural protein 13 (NSP13), which is thought to antagonize interferon signaling [93]. Finally, *RNF5* has been implicated in both promoting and antagonizing severe acute respiratory syndrome coronavirus 2 (SARS-CoV-2) by either stabilizing the interactions of membrane protein (M) [94] or inducing degradation of structural protein envelope (E) [95], respectively. Fundamentally, *RNF5* regulates virus-triggered interferon signaling by targeting the stimulator of interferon genes (STING) or mitochondrial antiviral signaling protein (MAVS) for ubiquitin-mediated protein degradation [73,74]. These studies reinforce the roles of tested candidate regulators in virus-associated processes, including interferon-mediated signaling.

## Future considerations for the use of trans-eQTL analysis in identification of L1 regulators

While we believe this approach can readily be applied to other datasets, we would like to note potential further considerations with the approach implemented here, some of which were simply beyond the scope of this paper. Firstly, though it is common to use probabilistic estimation of expression residuals (PEER) [40] to enhance detection of *cis*-eQTLs, PEER was not implemented in our main analyses as a precautionary measure, in order to avoid potentially blurring global TE signals, which likely led to a more conservative list of candidate *cis* gene mediators. Second, given the technical complexity in generating the vast amount of mRNA-seq data used for the eQTL analysis, it is possible that technical covariates introduced non-linear effects that would not be easily removed by approaches like PEER or SVA [96]. For that reason, we opted to supplement our computational predictions with experimental data. Third,

the L1 *trans*-eQTLs identified were specific to older L1 subfamilies (L1P and L1M) and were not shared across subfamilies. One factor that may partially explain this is the heightened difficulty of quantifying the expression of evolutionarily younger L1 subfamilies using short-read sequencing [97].

More generally, significant single gene differences are often difficult to reproduce across studies, and it is for this reason that methods like GSEA were developed, to robustly identify broader changes in sets of genes [45]. Consistently, GSEA suggests that many TE families, beyond the single L1 subfamilies identified in the eQTL analysis, are differentially regulated among samples with different genotypes for *trans*-eQTL SNVs and among samples where *IL16*/IL16 and *STARD5* were manipulated. We note that although *HLA* and *HSD17B12* loci were significant in both the European and African cohorts, we were not able to independently identify all of the same candidate regulators. This is likely due to a combination of small sample size for the African cohort and the existence of population-specific L1 regulation. Future studies with larger sample sizes may be useful for expanding the catalogue of loci that are biologically meaningful for L1 expression across more than one population. Importantly, our computational scan is limited to loci exhibiting genomic variation among tested individuals. This will vary with factors like the ancestry groups of the populations being studied. Moreover, variants that confer extreme fitness defects may not exist at a sufficiently high level in a population to allow for an assessment of their involvement as eQTLs.

Finally, although we focused on protein-coding candidate regulators, it is possible that the non-coding genes identified in our scan may also causally drive differences in L1 expression. Though not explored here, other regulatory factors like small RNAs may also act as partial mediators. Since the GEUVADIS Consortium generated small RNA data in parallel to the mRNA data used in this study [26], a future application of our pipeline could be to scan for *cis* small RNA mediators in the same biological samples. These unexplored factors may explain the associations between orphan SNV genotypes and TE family gene set changes.

### L1 trans-eQTLs are enriched for genetic variants linked to aging and age-related disease

Consistent with the notion that L1 is associated with aging and aging phenotypes [5,75], we observed that L1 *trans*-eQTL SNVs were associated with aging phenotypes in GWAS/PheWAS databases. This is very surprising, but interesting, given that all 1000Genomes Project participants declared themselves to be healthy at the time of sample collection. Assuming this to be true, our results suggest that L1 RNA level differences exist in natural, healthy human populations, and these RNA level differences precede onset of aging diseases. Importantly, we note that the SNVs tested were associated with intronic and nearby intergenic L1 subfamily RNA levels, which may have been discarded in studies focusing on full-length, distal intergenic L1 elements. Thus, these results reiterate the notion that intronic L1s and intergenic L1s near genes can potentially exert functional consequences on hosts and therefore merit further study. Though it is often unclear whether L1 mis-regulation is a consequence or driver of aging phenotypes, our results suggest that L1 RNA levels may drive aging phenotypes. As we continue to expand the catalogue of L1 regulators, especially in healthy cells and tissues, the L1 regulatory processes that are disrupted over the course of aging will become increasingly clear. To that end, this work may serve as a guide for conducting more comprehensive scans for candidate TE regulators.

In summary, we developed an eQTL-based pipeline that leverages genomic and transcriptomic data to scan the human genome for novel candidate regulators of L1 subfamily RNA levels. Though the initial scan identified genetic variants associated with RNA level differences in

specific L1 subfamilies, secondary analyses by GSEA suggest that genetic variants are associated with subtle but global differences in the RNA levels of many TE families. Our pipeline identified candidate genes, including *HSD17B12* and *HLA* genes, that likely play a conserved role in L1 regulation across human populations of different ancestries. Though some top candidates from the European cohort scan, such as *IL16*, *STARD5*, and *RNF5*, were not significant in the African cohort analysis, it is likely that some of these genes would appear in cross-ancestry scans with larger samples sizes. We detected subtle but global differences in L1 family RNA levels following *IL16* overexpression, *STARD5* overexpression, and rhIL16 treatment for 24 hours, further suggesting that some candidate genes have regulatory potential. We generate a list of pathways, such as mTORC1 signaling and cholesterol metabolism, that may act upstream of L1 regulation. Finally, the co-association of some genetic variants with both L1 RNA level differences and various age-related diseases suggests that L1 differences may precede and contribute to the onset of disease. Our results expand the potential mechanisms by which L1 RNA levels are regulated and by which L1 may influence aging-related phenotypes.

## Materials and methods

### Ethics statement

All animals were treated and housed in accordance with the Guide for Care and Use of Laboratory Animals. All experimental procedures were approved by the University of Southern California's Institutional Animal Care and Use Committee (IACUC) and are in accordance with institutional and national guidelines. Samples were derived from animals on approved IACUC protocol #20770.

### Publicly available data acquisition

The eQTL analysis was carried out on 358 European (EUR) individuals and 86 Yoruban (YRI) individuals for which paired single nucleotide variant, structural variant, and transcriptomic data were available from Phase 3 of the 1000Genomes Project [22,23] and from the GEUVADIS consortium [26]. Specifically, Phase 3 autosomal SNVs called on the GRCh38 reference genome were obtained from The International Genome Sample Resource (IGSR) FTP site (http://ftp.1000genomes.ebi.ac.uk/vol1/ftp/data_collections/1000_genomes_project/release/20190312_biallelic_SNV_and_INDEL/). Structural variants were also obtained from the IGSR FTP site (http://ftp.1000genomes.ebi.ac.uk/vol1/ftp/phase3/integrated_sv_map/). mRNA-sequencing fastq files generated by the GEUVADIS consortium were obtained from ArrayExpress under accession E-GEUV-1.

### Aggregating and pre-processing genotype data for eQTL analyses

To prepare SNVs for association analyses, all SNVs were first annotated with rsIDs from dbSNP build 155 using BCFtools v1.10.2 [98]. VCFtools v0.1.17 [99] was then used to remove indels and keep variants with the following properties in each of the two populations: possessed a minimum and maximum of two alleles, possessed a minor allele frequency (MAF) of at least 1%, passed Hardy-Weinberg equilibrium thresholding at p < 1e-6, with no missing samples, and located on an autosome. We note here that sex chromosomes were not included in the analysis since (i) Y chromosome SNVs were not available and (ii) analyses with X chromosome SNVs require unique algorithms and cannot simply be incorporated into traditional association pipelines [100,101]. VCF files containing these filtered SNVs were then converted to PLINK BED format using PLINK v1.90b6.17 [102], keeping the allele order. PLINK BED files were subsequently used to generate preliminary 0/1/2 genotype matrices using the '—

recodeA' flag in PLINK. These preliminary matrices were manipulated in terminal, using the gcut v9.0 function to remove unnecessary columns and datamash v1.7 to transpose the data, to generate the final 0/1/2 matrices used for the eQTL analyses. Finally, PLINK was used to prune the list of filtered SNVs, using the "—indep-pairwise 50 10 0.1" flag, and to generate principal components (PCs) from the pruned genotypes.

To control for inter-individual differences in genomic transposon copy number load, we applied 1 of 2 approaches, depending on the analysis. For approach 1, the net number of L1 and Alu insertions was quantified across the 444 samples. We chose to aggregate the L1 and Alu copy numbers, since Alu relies on L1 machinery for mobilization [103], and so the aggregate number may provide a finer view of L1-associated copy number load. Briefly, VCFTools was used to extract autosomal structural variants from the 1000Genomes structural variant calls. L1 and Alu insertions and deletions were then extracted with BCFtools by keeping entries with the following expressions: 'SVTYPE = "LINE1"', 'SVTYPE = "ALU"', 'SVTYPE = "DEL_LINE1"', and 'SVTYPE = "DEL_ALU"'. The resulting VCF files were then transformed to 0/1/2 matrices in the same manner as the SNVs. A net copy number score was obtained for each sample by adding the values for the L1 and Alu insertions and subtracting the values for the L1 and Alu deletions. For approach 2, the complete structural variant matrix was filtered with VCFtools using the same parameters as with the SNV matrices. The filtered structural variant matrix was then pruned with PLINK, and these pruned structural variant genotypes were used to generate principal components, in the same fashion as with the SNV matrix. The net copy number score or the structural variant principal components, depending on the analysis, were included as covariates.

## mRNA-seq read trimming, mapping, and quantification

Fastq files were first trimmed using fastp v0.20.1 [104] with the following parameters: detect_adapter_for_pe, disable_quality_filtering, trim_front1 17, trim_front2 17, cut_front, cut_front_window_size 1, cut_front_mean_quality 20, cut_tail, cut_tail_window_size 1, cut_tail_mean_quality 20, cut_right, cut_right_window_size 5, cut_right_mean_quality 20, and length_required 36. Read quality was then inspected using fastqc v0.11.9.

Next, the GRCh38 primary human genome assembly and comprehensive gene annotation were obtained from GENCODE release 33 [105]. Since LCLs are generated by infecting B-cells with Epstein-Barr virus, the EBV genome (GenBank ID V01555.2) was included as an additional contig in the human reference genome. The trimmed reads were aligned to this modified reference genome using STAR v2.7.3a [106] with the following parameters: outFilterMultimapNmax 100, winAnchorMultimapNmax 100, and outFilterMismatchNoverLmax 0.04. The TEcount function in the TEtranscripts v2.1.4 [38] package was employed to obtain gene and TE counts, using the GENCODE annotations to define gene boundaries and a repeat GTF file provided on the Hammell lab website (downloaded on February 19 2020 from https://labshare.cshl.edu/shares/mhammelllab/www-data/TEtranscripts/TE_GTF/GRCh38_GENCODE_rmsk_TE.gtf.gz) to define repeat boundaries.

Similarly, the TElocal v1.1.1 package (https://github.com/mhammell-laboratory/TElocal), from the same software suite as TEtranscripts, was employed to obtain gene and TE locus-specific counts using the same GENCODE annotations and a repeat file provided on the Hammell lab website (downloaded on October 31 2023 from https://labshare.cshl.edu/shares/mhammelllab/www-data/TElocal/prebuilt_indices/).

## Gene *cis*-eQTL and L1 *trans*-eQTL analyses

Gene and TE count files were loaded into R v4.2.1. Lowly expressed genes were first filtered out if 323/358 European samples and 78/86 Yoruban samples did not have over 0.44 counts

per million (cpm) or 0.43 cpm, respectively. These fractions were selected because they corresponded to expression in ~90% of samples and thus helped maintain maximal statistical power by focusing on genes ubiquitously expressed across each entire population. The cpm thresholds were selected because they corresponded to 10 reads in the median-length library within each set of samples. For the locus-specific quantifications, repeat counts were loaded into R and stratified into the following categories: (i) 'distal intergenic' TEs that were >5 kb from a gene, (ii) 'nearby intergenic' TEs that were within 5 kb of a gene, (iii) 'exonic' TEs that overlapped any annotated exon, and (iv) 'intronic' TEs for TEs that were in a gene but did not overlap an annotated exon. The stratification was carried out in order to separately characterize the influences and responses of each TE type to our analytical groups. After stratifying, repeat counts were aggregated at the subfamily level in order to compare results with the unstratified TEtranscripts results. After aggregating, lowly expressed genes were filtered as specified above.

Then, counts underwent a variance stabilizing transformation (vst) using DESeq2 v1.36.0 [107]. The following covariates were regressed out from vst normalized expression data using the 'removeBatchEffect' function in Limma v3.52.2 [108]: lab, population category, principal components 1–2 of the pruned SNVs, biological sex, net L1/Alu copy number, and EBV expression levels. Since the Yoruban samples were all from the same population, the population variable was omitted in their batch correction. Here, we note several things. First, EBV expression was included as a covariate because heightened TE expression is often a feature of viral infections [109]. Secondly, although PEER [40] is often used to remove technical variation for *cis*-eQTL analysis, this can come at the expense of correcting out genome-wide biological effects. This can be problematic in some settings, such as *trans*-eQTL analysis. Thus, PEER factors were not included in the main analysis. However, in a supplemental analysis, 10 PEER factors, in addition to the previously listed known covariates, were also regressed out from the gene expression profiles to determine whether unknown factors could explain the associations between top loci and L1 RNA levels (see below). The batch-corrected data in both cases underwent a final inverse normal transformation (INT), using the RankNorm function in the R package RNOmni v1.0.1, to obtain normally distributed gene expression values.

The INT expression matrices were split into genes and L1 subfamilies, which were used to independently identify gene *cis*-eQTLs and L1 subfamily *trans*-eQTLs in the European superpopulation using MatrixEQTL v2.3 [110]. For gene *cis*-eQTLs, SNVs were tested for association with expressed genes within 1 million base pairs. We opted to use a *trans*-eQTL approach using aggregate subfamily-level TE expression since the *trans* approach should allow us to identify regulators of many elements rather than one. The Benjamini-Hochberg false discovery rate (FDR) was calculated in each analysis, and we used the p-value corresponding to an FDR of < 5% as the threshold for eQTL significance. In addition, the *cis*-eQTL and *trans*-eQTL analyses were also repeated using 20 permuted expression datasets in which the sample names were scrambled, and the p-value corresponding to an average empirical FDR of < 5% was used as a secondary threshold. To note, we calculated the average empirical FDR at a given p-value $p_i$ by (i) counting the total number of null points with $p \leq p_i$, (ii) dividing by the number of permutations, to obtain an average number of null points with $p \leq p_i$, and (iii) dividing the average number of null points with $p \leq p_i$ by the number of real points with $p \leq p_i$. eQTLs were called as significant if they passed the stricter of the two thresholds. SNV-gene and SNV-L1 associations that were significant in the European superpopulation were then targeted and tested in the Yoruban population using R's built-in linear modelling functions. In this case, only the Benjamini-Hochberg FDR was calculated, and significant eQTLs were called if they possessed an FDR < 0.05.

### Defining SNV-gene-L1 trios and mediation analysis

For each population, the independent *cis-* and *trans-*eQTL results were integrated to identify SNVs associated with both gene and L1 subfamily expression. We reasoned that L1 expression would respond to differences in expression of *bona fide* regulators. Consequently, gene expression and L1 subfamily expression associations were assessed by linear regression, and the p-values from this analysis were Benjamini-Hochberg FDR-corrected. Candidate SNV-gene-L1 trios were defined as those with *cis-*eQTL, *trans-*eQTL, and expression regression FDRs < 0.05. To identify top, index SNVs in regions of linkage disequilibrium (LD), SNVs within 500 kilobases of each other with an $R^2 > 0.10$ were clumped together by *trans-*eQTL p-value using PLINK v1.90b6.17. To determine whether index SNVs were in regions enriched for L1 fragments, we used the Hammell lab repeat annotation file to identify the number of L1 fragments within 5 kb on either side of each index SNV or 1000 random SNVs. We assessed the statistical significance of the difference with a two-sample Wilcoxon test. Mediation analysis was carried out using the 'gmap.gpd' function in eQTLMAPT v0.1.0 [111] on all candidate SNV-gene-L1 trios. Empirical p-values were calculated using 30,000 permutations, and Benjamini-Hochberg FDR values were calculated from empirical p-values. Mediation effects were considered significant for trios with FDR < 0.05.

### Differential expression analysis across *trans*-eQTL SNV genotypes

Transcriptomic changes associated with alternating the allele of each SNV of interest were evaluated using DESeq2 v1.36.0. Using the same filtered counts prepared for the eQTL analysis, a linear model was constructed with the following covariates for each SNV: SNV genotype in 0/1/2 format, biological sex, lab, population category, principal components 1–2 of the pruned SNVs, and principal components 1–3 of the pruned SVs (to account for structural variant population structure). As before, the population label was omitted from the Yoruban population analysis. Significant genes and TEs were those with an FDR < 0.05.

### Functional enrichment analyses

We used the Gene Set Enrichment Analysis (GSEA) paradigm as implemented in the R package clusterProfiler v4.4.4 [112]. Gene Ontology, Reactome, and Hallmark pathway gene sets were obtained from the R package msigdbr v7.5.1, an Ensembl ID-mapped collection of gene sets from the Molecular Signature Database [45,49]. Additionally, TE subfamilies were aggregated into TE family gene sets using the TE family designations specified in the TE GTF file (downloaded on February 19 2020 from https://labshare.cshl.edu/shares/mhammelllab/www-data/TEtranscripts/TE_GTF/GRCh38_GENCODE_rmsk_TE.gtf.gz) or index file (downloaded on October 31 2023 from https://labshare.cshl.edu/shares/mhammelllab/www-data/TElocal/prebuilt_indices/) used during the RNA-seq quantification step. The DESeq2 v1.36.0 Wald-statistic was used to generate a combined ranked list of genes and TEs for functional enrichment analysis. All gene sets with an FDR < 0.05 were considered significant. For plots with a single analysis, the top 5 downregulated and top 5 upregulated gene sets were plotted, at most. For plots with multiple analyses, shared gene sets with the desired expression patterns in each individual analysis were first identified. Then, the p-values for shared gene sets were combined using Fisher's method, and this meta-analysis p-value was used to rank shared gene sets. Finally, the top 5 gene sets with one expression pattern and the top 5 gene sets with the opposite expression pattern were plotted. If there were less than 5 gene sets in either group, those were replaced with gene sets exhibiting the opposite regulation, in order to plot 10 shared gene sets whenever possible.

## Cell lines and cell culture conditions

GM12878 (RRID: CVCL_7526) lymphoblastoid cells were purchased from the Coriell Institute. We opted to use GM12878 as a well-characterized representative cell line for candidate validation, given that (i) it is of the same cell type as the transcriptomic data used here for our eQTL analysis, and (ii) its epigenomic landscape and culture conditions are well-characterized as part of the ENCODE project [56,57].

GM12878 cells were maintained in RPMI (Corning cat. 15-040-CV) containing 15% FBS and 1X Penicillin-Streptomycin-Glutamine (Corning cat. 30-009-CI). Cells were cultured in a humidified incubator at 37˚C and 5% $CO_2$, subculturing cells 1:5 once cells reached a density of ~$10^6$ mL$^{-1}$. All cells used were maintained below passage 30 and routinely tested for mycoplasma contamination using the PlasmoTest Mycoplasma Detection Kit (InvivoGen).

## Plasmids

The empty pcDNA3.1(+) backbone (Invitrogen cat. V79020) was a kind gift from the lab of Dr. Changhan David Lee at the University of Southern California Leonard Davis School of Gerontology. Overexpression vectors for *IL16* (CloneID OHu48263C), *STARD5*-FLAG (CloneID OHu07617D), *HSD17B12*-FLAG (CloneID OHu29918D), and *RNF5*-FLAG (CloneID OHu14875D) on a pcDNA3.1 backbone were purchased from GenScript. Plasmid sequences were verified for accuracy using Plasmidsaurus's whole plasmid sequencing service.

## Transfections

*Escherichia coli* were cultured in LB Broth (ThermoFischer Scientific) supplemented with 50 μg/mL carbenicillin to an optical density 600 ($OD_{600}$) of 2–4. Plasmid extractions were carried out using the Nucleobond Xtra Midi Plus EF kit (Macherey-Nagel) following manufacturer recommendations. Plasmids were aliquoted and stored at -20˚C until the time of transfection. On the day of transfection, GM12878 cells were collected in conical tubes, spun down (100xG, 5 minutes, room temperature), resuspended in fresh media, and counted by trypan blue staining using a Countess II FL automated cell counter (Thermo Fisher). The number of cells necessary for the experiment were then aliquoted, spun down, and washed with Dulbecco's phosphate-buffered saline (DPBS)(Corning, cat. #21-031-CV).

GM12878 cells were transfected by electroporation using the Neon Transfection System (Invitrogen) with the following parameters: 1200 V, 20 ms, and 3 pulses for GM12878 cells in Buffer R. Per reaction, we maintained a plasmid mass:cell number ratio of 10 μg: $2*10^6$ cells. For mRNA-sequencing, $8*10^6$ GM12878 cells were independently transfected for each biological replicate, with 4 replicates per overexpression condition, and cultured in a T25 flask. Immediately after transfection, cells were cultured in Penicillin-Streptomycin-free media for ~24 hours.

Afterwards, to promote selection of viable and healthy transfected GM12878 cells, we enriched for viable cells using the EasySep Dead Cell Removal (Annexin V) Kit (STEMCELL Technologies) before seeding $2*10^6$ live cells in the same media used for cell maintenance. After another 24 hours, cell viability was measured by trypan blue staining on a Countess automated cell counter and cells were spun down (100xG, 5 min, room temperature) and lysed in TRIzol Reagent (Invitrogen) for downstream total RNA isolation (see below).

## Recombinant human IL16 (rhIL16) peptide treatment

Human rIL16 was obtained from PeproTech (cat. #200–16) and resuspended in 0.1% bovine serum albumin (BSA) solution (Akron, cat. #AK8917-0100). GM12878 cells were seeded at a

concentration of 500,000 live cells per mL of media on 6-well suspension plates with 3 independent replicates per condition. Cells were exposed to 0, 24, or 48 hours of 100 ng mL$^{-1}$ of rhIL16. To replace or exchange media 24 hours after seeding, cells were transferred to conical tubes, spun down (100xG, 5 min, room temperature), resuspended in 5 mL of the appropriate media, and transferred back to 6-well suspension plates. After 48 hours, cell viability was measured by trypan blue staining and cells were spun down (100xG, 5 min, room temperature) and lysed in TRIzol Reagent (Invitrogen).

### RNA extractions and mRNA sequencing

RNA was extracted using the Direct-zol RNA Miniprep kit (Zymo Research) following manufacturer recommendations. The integrity of RNA samples was evaluated using an Agilent High Sensitivity RNA ScreenTape assay (Agilent Technologies), ensuring that all samples had a minimum eRIN score of 8 before downstream processing. We then submitted total RNA samples to Novogene (Sacramento, California) for mRNA library preparation and sequencing on the NovaSeq 6000 platform as paired-end 150 bp reads.

### Analysis of overexpression and rhIL16 exposure mRNA-seq

mRNA-seq reads were trimmed, mapped, and quantified like for the eQTL analysis, except for the overexpression sample data. For this data, one modification was made: the EBV-inclusive reference genome was further modified to include the pcDNA3.1 sequence as an additional contig. Lowly expressed genes were filtered using a cpm threshold as in the eQTL processing, but that cpm threshold had to be satisfied by as many samples as the size of the smallest biological group. For the overexpression data, surrogate variables were estimated with the 'svaseq' function [96] in the R package 'sva' v3.44.9, and they were regressed out from the raw read counts using the 'removeBatchEffect' function in the R package Limma v3.52.2. DESeq2 was used to identify significantly (FDR < 0.05) differentially expressed genes and TEs between groups. Functional enrichment analysis was carried out as previously described.

### PheWAS analysis

To gather the known associated traits for the 499 TE-related SNVs, we used Open Targets Genetics (https://genetics.opentargets.org/), a database of GWAS summary statistics [113]. First, we queried the database using the 499 TE-related SNVs and collected traits that were directly associated (with $P < 5x10^{-8}$) with the SNVs, as well as traits associated with lead variants that were in linkage disequilibrium (LD) with the queried SNPs (with $R^2 > 0.6$). For age-related traits (ARTs), we used the comprehensive list of 365 Medical Subject Headings (MeSH) terms reported by [114] (downloaded from https://github.com/kisudsoe/Age-related-traits). To identify known age-related traits, the known associated traits were translated into the equivalent MeSH terms using the method described by [114]. Then, the MeSH-translated known associated traits for the 499 TE-related SNVs were filtered by the MeSH terms for age-related traits.

As a parallel approach, we mapped the RsIDs for all SNVs used during the eQTL analyses to their corresponding bi-allelic Open Targets variant IDs, when available. The variant IDs corresponding to L1 *trans*-eQTL SNVs were extracted, and 500 different equal-length combinations of random SNVs were generated. Next, we queried the Open Targets database using the lists of L1-associated and random SNVs and collected the associated traits (with $P < 5x10^{-8}$). Importantly, the database assigns traits to broader categories, including 14 disease categories that we considered age-related. We counted the number of L1-associated or random SNVs mapping to each category, and we used the random SNV counts to generate an empirical

cumulative distribution function (ecdf) for each category. We calculated enrichment p-values using the formula p = 1- ecdf(mapped eQTLs) and then Benjamini-Hochberg FDR-corrected all p-values. An enrichment score (ES) was also calculated for each category using the formula ES = number of mapped L1 eQTLs / median number of randomly mapping SNVs. Categories with an ES > 1 and FDR < 0.05 were considered significantly enriched.

### Quantification of mouse serum IL16 by ELISA

Serum was collected from male and female C57BL/6JNia mice (4–6 and 20–24 months old) obtained from the National Institute on Aging (NIA) colony at Charles Rivers. All animals were euthanized between 8–11 am in a "snaking order" across all groups to minimize batch-processing confounds due to circadian processes. All animals were euthanized by $CO_2$ asphyxiation followed by cervical dislocation. Circulating IL16 levels were quantitatively evaluated from mouse serum by enzyme-linked immunosorbent assay (ELISA). Serum was diluted 1/10 before quantifying IL16 concentrations using Abcam's Mouse IL-16 ELISA Kit (ab201282) in accordance with manufacturer instructions. Technical replicates from the same sample were averaged to one value before statistical analysis and plotting. P-values across age within each sex were calculated using a non-parametric 2-sided Wilcoxon test, and p-values from each sex-specific analysis were combined using Fisher's method.

### Additional software versions

Analyses were conducted using R version 4.2.1. Code was re-run independently on R version 4.3.0 to check for reproducibility. Since PEER is no longer maintained, a Docker image containing R version 3.4.1 and PEER version 1.3 [quay.io/biocontainers/r-peer:1.3—r341h470a237_1] were used for the sole purpose of identifying PEER factors for a supplemental L1 *trans*-eQTL analysis.

### Supporting information

**S1 Fig. Quality control of GEUVADIS and 1000Genomes data. (A)** Unique and multimapping fractions for the mRNA-sequencing data, color-coded by ancestry group, laboratory, or biological sex. For visual clarity, the plots were limited to samples with a minimum of 84% uniquely mapped reads, a limit which still captured the majority of samples analyzed. For ancestry, the following groups were analyzed: Tuscan (TSI), Northern Europeans from Utah (CEU), Finnish (FIN), British (GBR), and Yoruba (YRI). **(B)** PCA plots for pruned SNV genotype data from European or African samples, color-coded and shaped according to ancestry group. **(C)** PCA plots for pruned SV genotype data from European or African samples, color-coded and shaped according to ancestry group.
(TIF)

**S2 Fig. Annotation of genome-wide SNVs with *cis*-eQTLs in the European cohort. (A)** A Manhattan plot for the gene *cis*-eQTL analysis in the European cohort. The dashed line at p = 4.75E-4 corresponds to an average empirical FDR < 0.05, based on 20 random permutations. One such permutation is illustrated in the bottom panel. The solid line at p = 4.73E-4 corresponds to a Benjamini-Hochberg FDR < 0.05. The stricter of the two thresholds, p = 4.73E-4, was used to define significant *cis*-eQTLs. **(B)** EBV expression as a function of *IL16* expression prior to (left) and after (right) correcting for EBV expression. This correction was applied prior to the eQTL scan in order to avoid confounding SNV associations with differences in EBV expression. **(C)** A Manhattan plot for the L1 subfamily *trans*-eQTL analysis in the European cohort, after correcting for known covariates as well as 10 PEER factors. The

genes and L1 subfamilies identified in the original analysis without PEER are listed next to the equivalent peaks in this analysis. The solid line at p = 2.44E-7 corresponds to a Benjamini-Hochberg FDR < 0.05, which was used to define significant *trans*-eQTLs. FDR: False Discovery Rate.
(TIF)

**S3 Fig. Identification of 2[nd] tier candidate L1 RNA level regulators in the European cohort.** **(A)** A schematic for how 2[nd] tier candidate genes were defined. In short, these were genes in trios with clumped SNVs but not index SNVs at the top of each peak. **(B)** The three-part integration results for three genes—*RNF5, EHMT2-AS1, FKBPL*—that we considered second tier candidates for functional, *in vitro* testing. In the left column are the *trans*-eQTLs, in the middle column are the *cis*-eQTLs, and in the right column are the linear regressions for gene expression against L1 subfamily RNA levels. Expression values following an inverse normal transform (INT) are shown. The FDR for each analysis is listed at the top of each plot. **(C)** The number of L1 fragments near 1000 random SNVs or the 5 *trans*-eQTL index SNVs were calculated. A two-sample Wilcoxon test was run to determine whether there were significant differences between the two groups. FDR: False Discovery Rate.
(TIF)

**S4 Fig. 1[st] and 2[nd] tier candidate regulators are partial mediators of SNV effects on L1 RNA levels. (A)** Scheme for the mediation analysis. Mediation analysis tests the mechanistic model where a given gene, *in cis* to a given SNV, partially or fully mediates the effect that SNV has on TE RNA levels *in trans*. The direct, indirect, and total effects for **(B)** 1st tier candidate gene trios and **(C)** 2[nd] tier candidate gene trios are shown. Mediation was considered significant if the FDR-adjusted empirical p-value, calculated from 30,000 permutations, was < 5%. FDR: False Discovery Rate.
(TIF)

**S5 Fig. *In silico* scanning for candidate L1 RNA level regulators in an independent, African population. (A)** A scheme for the *in silico* analysis carried out with an African cohort, which was similar to the analysis with the European cohort. Since the sample size was smaller than the European cohort, a targeted eQTL approach was undertaken, where we 1) only checked for replication of significant L1 *trans*-eQTLs and 2) only tested significant *trans*-eQTL SNVs for *cis*-association with genes that were L1-linked in the European cohort analysis. **(B)** The three-part integration results for two 1[st] tier candidate regulators in the African cohort—*HSD17B12* and *HLA-DRB6*. In the left column are the *trans*-eQTLs, in the middle column are the *cis*-eQTLs, and in the right column are the linear regressions for gene expression against L1 subfamily expression. Expression values following an inverse normal transform (INT) are shown. The FDR for each analysis is listed at the top of each plot. FDR: False Discovery Rate. Panel **(A)** was created with BioRender.com.
(TIF)

**S6 Fig. Some European-derived candidate TE-regulator genes do not replicate in the African cohort.** Many candidate genes from the *in silico* screen in the European cohort did not replicate in the African cohort, likely due to the much smaller sample size and relative rarity of homozygotes carrying 2 alternate alleles. **(A)** The three-part integration results for 1[st] and 2[nd] tier candidate regulators identified in the European cohort analysis and tested in the African cohort. In the left column are the *trans*-eQTLs, in the middle column are the *cis*-eQTLs, and in the right column are the linear regressions for gene expression against L1 subfamily expression. Expression values following an inverse normal transform (INT) are shown. The FDR is listed for each integration step except for the linear regressions since trios were filtered out

before the regression step. FDR: False Discovery Rate.
(TIF)

**S7 Fig. Scanning for candidate regulators of intronic, intergenic, or exonic L1 RNA levels.** Transposon locus-specific quantifications were obtained using the TElocal package, and these were stratified by genomic region (i.e. intronic, nearby intergenic for loci < 5 kb from a gene, distal intergenic for loci > 5 kb from a gene, and exon-overlapping). Counts were then aggregated at the subfamily level, and the L1 eQTL scan was re-run using each of the four L1 expression profiles using the European cohort. The Manhattan plots for the **(A)** intronic L1 subfamily, **(B)** nearby intergenic L1 subfamily, **(C)** distal intergenic L1 subfamily, and **(D)** exon-overlapping L1 subfamily *trans*-eQTL analyses. For readability, only a subset of associated genes and L1s are highlighted in each plot. For regions with *trans*-eQTLs for multiple L1 subfamilies, we used an un-pointed line to depict the association between the listed L1 subfamilies and at least one SNV in that region. The solid red line in each plot corresponds to a Benjamini-Hochberg FDR < 0.05. FDR: False Discovery Rate.
(TIF)

**S8 Fig. Identifying mediators of SNV effects on distal intergenic L1 RNA levels. (A)** Scheme for the mediation analysis. Mediation analysis tests the mechanistic model where a given gene, *in cis* to a given SNV, partially or fully mediates the effect that SNV has on TE expression *in trans*. **(B)** The three-part integration results for one protein-coding gene—*ZSCAN26*—that we considered a candidate regulator of distal intergenic L1 RNA levels. In the left column are the *trans*-eQTLs, in the middle column are the *cis*-eQTLs, and in the right column are the linear regressions for gene expression against L1 subfamily RNA levels. Expression values following an inverse normal transform (INT) are shown. The FDR for each analysis is listed at the top of each plot. **(C)** The *ZSCAN26* SNV-gene-TE mediation trio results. Mediation was considered significant if the FDR-adjusted empirical p-value, calculated from 30,000 permutations, was < 5%. FDR: False Discovery Rate.
(TIF)

**S9 Fig. Total L1 *trans*-eQTLs alter the expression of TEs in distinct genomic regions.** Box and whisker plots for the $\log_2$ fold changes of TE subfamilies (red dots), grouped by TE family, across genotypes for **(A)** rs11635336 (*IL16/STARD5*), **(B)** rs9271894 (*HLA*), and **(C)** rs1061810 (*HSD17B12*). A one-sample Wilcoxon test was run to determine whether changes were significantly different from 0. The FDR values from this test are listed at the bottom. GSEA analysis for top, differentially regulated TE family gene sets in different genomic regions (intronic, intergenic, exon-overlapping) across genotypes. The results for **(D)** rs11635336 (*IL16/STARD5*), **(E)** rs9271894 (*HLA*), and **(F)** rs1061810 (*HSD17B12*) are shown. In each bubble plot, the size of the dot represents the $-\log_{10}$(FDR) and the color reflects the normalized enrichment score. FDR: False Discovery Rate.
(TIF)

**S10 Fig. L1 *trans*-eQTL orphan SNVs are associated with differences in TE families and TE-associated pathways. (A)** Scheme for functionally annotating orphan index SNVs by GSEA. **(B)** GSEA analysis for shared, significantly regulated TE family gene sets across genotypes for rs112581165 and rs72691418. **(C)** GSEA plots for the L1 family gene set results summarized in **(B)**. For these plots, the FDR value is listed. **(D)** GSEA analysis for shared, significantly regulated, evolutionary-age-stratified L1 gene sets across genotypes for rs112581165 and rs72691418. L1M subfamilies are the oldest, L1P subfamilies are intermediate, and L1PA subfamilies are the youngest. GSEA analysis for top, shared, concomitantly regulated **(E)** MSigDB Hallmark pathway, **(F)** GO Biological Process, and **(G)** Reactome pathway

gene sets across genotypes for rs112581165 and rs72691418. Shared gene sets were ranked by combining p-values from each individual SNV analysis using Fisher's method. In each bubble plot, the size of the dot represents the -$\log_{10}$(FDR) and the color reflects the normalized enrichment score. FDR: False Discovery Rate.
(TIF)

**S11 Fig. L1 *trans*-eQTLs, in the absence of a known mediator, alter the levels of TEs in distinct genomic regions.** Box and whisker plots for the $\log_2$ fold changes of TE subfamilies (red dots), grouped by TE family, across genotypes for **(A)** rs112581165 and **(B)** rs72691418. A one-sample Wilcoxon test was run to determine whether changes were significantly different from 0. The FDR values from this test are listed at the bottom. GSEA analysis for top, differentially regulated TE family gene sets in different genomic regions (intronic, intergenic, exon-overlapping) across genotypes. The results for **(C)** rs112581165 and **(D)** rs72691418 are shown. In each bubble plot, the size of the dot represents the -$\log_{10}$(FDR) and the color reflects the normalized enrichment score. FDR: False Discovery Rate.
(TIF)

**S12 Fig. Distal intergenic L1 *trans*-eQTLs are associated with alterations in inflammatory pathways. (A)** Scheme for functionally annotating gene-linked index SNVs for distal intergenic L1 expression by GSEA. GSEA analysis for top regulated **(B)** MSigDB Hallmark pathway, **(C)** GO Biological Process, and **(D)** Reactome pathway gene sets across genotypes for rs1361387 (*ZSCAN26)*. In each bubble plot, the size of the dot represents the -$\log_{10}$(FDR) and the color reflects the normalized enrichment score. FDR: False Discovery Rate.
(TIF)

**S13 Fig. GM12878 LCLs as a model for assessing the roles of candidate genes in L1 regulation.** Though transcriptomic data is not available for these LCLs, the relative expression of candidate genes and linked L1 subfamilies can be predicted from GM12878 genotypes at either **(A)** index SNVs or **(B)** clumped SNVs. **(C)** ENCODE project epigenetic data available for GM12878 highlights regulatory markers near some *trans*-eQTL index SNVs. Data is visualized on the UCSC Genome Browser.
(TIF)

**S14 Fig. Though *HSD17B12* and *RNF5* were overexpressed, no L1 gene set changes were detected. (A)** Scheme for experimentally validating the roles of *IL16*, *STARD5*, *HSD17B12*, and *RNF5* in L1 regulation. **(B)** VST-normalized $\log_2$ counts were quantified by DESeq2 for each gene being overexpressed. Each dot represents an independent transfection, with n = 4 per condition. The FDR for each comparison is listed at the top. **(C)** VST-normalized $\log_2$ counts for EBV were quantified by DESeq2 for each condition. Each dot represents an independent transfection, with n = 4 per condition. The FDR for each comparison is listed at the top. **(D)** Expression heatmaps for the four candidate genes tested, under each overexpression condition. GSEA analysis for top, differentially regulated **(E)** GO Biological Process gene sets following *HSD17B12* overexpression. GSEA analysis for top, differentially regulated **(F)** GO Biological Process gene sets following *RNF5* overexpression. In each bubble plot, the size of the dot represents the -$\log_{10}$(FDR) and the color reflects the normalized enrichment score. FDR: False Discovery Rate. Panel **(A)** was created with BioRender.com.
(TIF)

**S15 Fig. *IL16* and *STARD5* overexpression alter the levels of TEs in distinct genomic regions.** Box and whisker plots for the $\log_2$ fold changes of TE subfamilies (red dots), grouped by TE family, following **(A)** *IL16* overexpression and **(B)** *STARD5* overexpression. A one-

sample Wilcoxon test was run to determine whether changes were significantly different from 0. The FDR values from this test are listed at the bottom. GSEA analysis for top, differentially regulated TE family gene sets in different genomic regions (intronic, intergenic, exon-overlapping) across overexpression condition. The results for **(C)** *IL16* overexpression and **(D)** *STARD5* overexpression are shown. In each bubble plot, the size of the dot represents the -$\log_{10}$(FDR) and the color reflects the normalized enrichment score. FDR: False Discovery Rate.
(TIF)

**S16 Fig. The effects of rhIL16 on an L1 family gene set are diminished after 48 hours. (A)** Scheme for experimentally validating the role of rhIL16 in L1 regulation. **(B)** Spearman rank correlation between (*top)* 24-hour rhIL16 treatment differential expression or (*bottom)* 48-hour rhIL16 treatment differential expression and *IL16* overexpression differential expression. **(C)** VST-normalized $\log_2$ counts for EBV were quantified by DESeq2 for each peptide treatment condition. Each dot represents an independent exposure, with n = 3 per condition. FDR > 0.05 at both the 24- and 48-hour treatment time points. **(D)** Box and whisker plots for the $\log_2$ fold changes of TE subfamilies (red dots) grouped by TE family after treatment with rhIL16 for 24 hours. A one-sample Wilcoxon test was run to determine whether changes were significantly different from 0. The FDR values from this test are listed at the bottom. GSEA analysis for top, shared, concomitantly regulated **(E)** GO Biological Process, **(F)** Reactome pathway, and **(G)** TE family gene sets following *IL16* overexpression, rhIL16 exposure for 24 hours, and rhIL16 exposure for 48 hours. Shared gene sets were ranked by combining p-values from each individual treatment analysis using Fisher's method. In each bubble plot, the size of the dot represents the -$\log_{10}$(FDR) and the color reflects the normalized enrichment score. FDR: False Discovery Rate. Panel **(A)** was created with BioRender.com.
(TIF)

**S17 Fig. Aging-related disease categories are enriched among L1 *trans*-eQTL PheWAS associations. (A)** Scheme for assessing enrichment of disease category associations among L1 *trans*-eQTLs. L1 eQTLs or combinations of random SNVs were queried on the Open Targets Genetics platform, and the number of SNVs mapping to 14 disease categories annotated by the platform were calculated. An empirical cumulative distribution function (ecdf) was defined for each disease category using the associations for the random SNVs, and this function was used to calculate an enrichment p-value, defined as p = 1 −ecdf(mapped eQTLs). Afterwards, p-values were FDR-corrected. An enrichment score (ES) was calculated by taking the number of mapped L1 eQTLs for a category and dividing by the median number of mapped SNVs among the random SNV combinations. Categories with an ES > 1 and FDR < 0.05 were considered significantly enriched. The number of mapped SNVs relative to the cumulative probabilities from the simulations are shown for **(B)** cell proliferation disorders, **(C)** diseases of the ear, **(D)** diseases of the visual system, **(E)** endocrine system diseases, **(F)** gastrointestinal diseases, **(G)** hematologic diseases, **(H)** immune system diseases, **(I)** integumentary system diseases, **(J)** musculoskeletal diseases, **(K)** nervous system diseases, **(L)** pancreas diseases, **(M)** respiratory diseases, **(N)** urinary system diseases, and **(O)** cardiovascular diseases. FDR: False Discovery Rate. Panel **(A)** was created with BioRender.com.
(TIF)

**S1 Table. Results for eQTL scans and mediation analysis in the European cohort. (A)** L1 subfamily *trans*-eQTLs passing FDR < 0.05 in the European populations. **(B)** Gene *cis*-eQTLs passing FDR < 0.05 in the European populations. **(C)** SNV-Gene-L1 trios passing FDR < 0.05 in the *cis*-eQTL, *trans*-eQTL, and linear regression analyses in the European cohort. **(D)** L1

subfamily *trans*-eQTLs passing Benjamini-Hochberg FDR < 0.05 in the European populations, after correcting for PEER factors. **(E)** European L1 *trans*-eQTLs clumped by p-value. **(F)** Significant SNV-Gene-L1 trios that survived clumping in the European populations. **(G)** Mediation analysis results for all SNV-Gene-L1 trios in the European cohort. FDR: False Discovery Rate.
(XLSX)

**S2 Table. Results for eQTL scans and mediation analysis in the African cohort. (A)** L1 subfamily *trans*-eQTLs passing FDR < 0.05 in the Yoruban population. **(B)** Gene *cis*-eQTLs passing FDR < 0.05 in the Yoruban population. **(C)** SNV-Gene-L1 trios passing FDR < 0.05 in the *cis*-eQTL, *trans*-eQTL, and linear regression analyses in the Yoruban population. **(D)** Yoruban L1 *trans*-eQTLs clumped by p-value. **(E)** Significant SNV-Gene-L1 trios that survived clumping in the Yoruban population. **(F)** Mediation analysis results for all SNV-Gene-L1 trios in the Yoruban cohort. FDR: False Discovery Rate.
(XLSX)

**S3 Table. Results for eQTL scans using intronic, intergenic, or exon-overlapping L1 subfamily RNA levels. (A)** Intronic L1 subfamily *trans*-eQTLs passing FDR < 0.05 in the EUR population. **(B)** Nearby intergenic L1 subfamily *trans*-eQTLs passing FDR < 0.05 in the EUR population. **(C)** Distal intergenic L1 subfamily *trans*-eQTLs passing FDR < 0.05 in the EUR population. **(D)** Exon overlapping L1 subfamily *trans*-eQTLs passing FDR < 0.05 in the EUR population. **(E)** Significant SNV-Gene-L1 trios for intronic L1 subfamilies. **(F)** Significant SNV-Gene-L1 trios for nearby intergenic L1 subfamilies. **(G)** Significant SNV-Gene-L1 trios for distal intergenic L1 subfamilies. **(H)** Significant SNV-Gene-L1 trios for exon-overlapping L1 subfamilies. **(I)** Mediation analysis results for distal intergenic L1-SNV-gene trios.
(XLSX)

**S4 Table. Results for the differential gene expression analysis and GSEA comparing genotypes in SNVs with an attributed *cis*-mediator. (A)** All DESeq2 results for alternating the allele of rs11635336. **(B)** All DESeq2 results for alternating the allele of rs9271894. **(C)** All DESeq2 results for alternating the allele of rs1061810. **(D)** All GSEA results for rs11635336 using TE family gene sets. **(E)** All GSEA results for rs9271894 using TE family gene sets. **(F)** All GSEA results for rs1061810 using TE family gene sets. **(G)** All GSEA results for rs11635336 using MSigDB Hallmark gene sets. **(H)** All GSEA results for rs9271894 using MSigDB Hallmark gene sets. **(I)** All GSEA results for rs1061810 using MSigDB Hallmark gene sets. **(J)** All GSEA results for rs11635336 using GO Biological Process gene sets. **(K)** All GSEA results for rs9271894 using GO Biological Process gene sets. **(L)** All GSEA results for rs1061810 using GO Biological Process gene sets. **(M)** All GSEA results for rs11635336 using Reactome gene sets. **(N)** All GSEA results for rs9271894 using Reactome gene sets. **(O)** All GSEA results for rs1061810 using Reactome gene sets. **(P)** Overlapping TE family gene sets for rs11635336, rs9271894, rs1061810. **(Q)** Overlapping age-stratified TE family gene sets for rs11635336, rs9271894, rs1061810. **(R)** All DESeq2 results for alternating the allele of rs9270493. **(S)** All GSEA results for rs9270493 using TE family gene sets. **(T)** GSEA results for genomic region-stratified TE family gene sets for rs11635336. **(U)** GSEA results for genomic region-stratified TE family gene sets for rs9271894. **(V)** GSEA results for genomic region-stratified TE family gene sets for rs1061810. **(W)** Overlapping MSigDB Hallmark gene sets for rs11635336, rs9271894, rs1061810. **(X)** Overlapping GO Biological Process gene sets for rs11635336, rs9271894, rs1061810. **(Y)** Overlapping Reactome gene sets for rs11635336, rs9271894, rs1061810.
(XLSX)

**S5 Table. Results for the differential gene expression analysis and GSEA comparing genotypes in orphan index SNVs and distal intergenic-associated SNVs.** (**A**) All DESeq2 results for alternating the allele of rs112581165. (**B**) All DESeq2 results for alternating the allele of rs72691418. (**C**) All GSEA results for rs112581165 using TE family gene sets. (**D**) All GSEA results for rs72691418 using TE family gene sets. (**E**) All GSEA results for rs112581165 using MSigDB Hallmark gene sets. (**F**) All GSEA results for rs72691418 using MSigDB Hallmark gene sets. (**G**) All GSEA results for rs112581165 using GO Biological Process gene sets. (**H**) All GSEA results for rs72691418 using GO Biological Process gene sets. (**I**) All GSEA results for rs112581165 using Reactome gene sets. (**J**) All GSEA results for rs72691418 using Reactome gene sets. (**K**) Overlapping TE family gene sets for rs112581165 and rs72691418. (**L**) Overlapping age-stratified TE family gene sets for rs112581165 and rs72691418. (**M**) GSEA results for genomic region-stratified TE family gene sets for rs112581165. (**N**) GSEA results for genomic region-stratified TE family gene sets for rs72691418. (**O**) Overlapping MSigDB Hallmark gene sets for rs112581165 and rs72691418. (**P**) Overlapping GO Biological Process gene sets for rs112581165 and rs72691418. (**Q**) Overlapping Reactome gene sets for rs112581165 and rs72691418. (**R**) All DESeq2 results for alternating the allele of rs1361387. (**S**) All GSEA results for rs1361387 using MSigDB Hallmark gene sets. (**T**) All GSEA results for rs1361387 using GO Biological Process gene sets. (**U**) All GSEA results for rs1361387 using Reactome gene sets. (XLSX)

**S6 Table. Results for the differential gene expression analysis and GSEA assessing the effects of overexpressing *IL16*, *STARD5*, *HSD17B12*, or *RNF5*.** (**A**) All DESeq2 results for *IL16* overexpression. (**B**) All DESeq2 results for *STARD5* overexpression. (**C**) All DESeq2 results for *HSD17B12* overexpression. (**D**) All DESeq2 results for *RNF5* overexpression. (**E**) All GSEA results for *IL16* overexpression using MSigDB Hallmark gene sets. (**F**) All GSEA results for *STARD5* overexpression using MSigDB Hallmark gene sets. (**G**) All GSEA results for *HSD17B12* overexpression using MSigDB Hallmark gene sets. (**H**) All GSEA results for *RNF5* overexpression using MSigDB Hallmark gene sets. (**I**) All GSEA results for *IL16* overexpression using GO Biological Process gene sets. (**J**) All GSEA results for *STARD5* overexpression using GO Biological Process gene sets. (**K**) All GSEA results for *HSD17B12* overexpression using GO Biological Process gene sets. (**L**) All GSEA results for *RNF5* overexpression using GO Biological Process gene sets. (**M**) All GSEA results for *IL16* overexpression using Reactome gene sets. (**N**) All GSEA results for *STARD5* overexpression using Reactome gene sets. (**O**) All GSEA results for *HSD17B12* overexpression using Reactome gene sets. (**P**) All GSEA results for *RNF5* overexpression using Reactome gene sets. (**Q**) All GSEA results for *IL16* overexpression using TE family gene sets. (**R**) All GSEA results for *STARD5* overexpression using TE family gene sets. (**S**) All GSEA results for *HSD17B12* overexpression using TE family gene sets. (**T**) All GSEA results for *RNF5* overexpression using TE family gene sets. (**U**) GSEA with age-stratified L1 family gene sets following *IL16* and *STARD5* OE. (**V**) GSEA results for genomic region-stratified TE family gene sets for *IL16* OE. (**W**) GSEA results for genomic region-stratified TE family gene sets for *STARD5* OE.
(XLSX)

**S7 Table. Results for the differential gene expression analysis and GSEA assessing the effects of rhIL16 exposure.** (**A**) All DESeq2 results for rhIL16 exposure for 24 hours. (**B**) All GSEA results for 24-hour rhIL16 exposure using MSigDB Hallmark gene sets. (**C**) All GSEA results for 24-hour rhIL16 exposure using GO Biological Process gene sets. (**D**) All GSEA results for 24-hour rhIL16 exposure using Reactome gene sets. (**E**) All GSEA results for 24-hour rhIL16 exposure using TE family gene sets. (**F**) Overlapping MSigDB Hallmark gene sets for *IL16* overexpression and rhIL16 24-hour exposure. (**G**) Overlapping GO Biological

Process gene sets for *IL16* overexpression and rhIL16 24-hour exposure. **(H)** Overlapping Reactome gene sets for *IL16* overexpression and rhIL16 24-hour exposure. **(I)** GSEA with age-stratified L1 family gene sets following rhIL16 24-hour exposure. **(J)** GSEA results for genomic region-stratified TE family gene sets for the 24-hour rhIL16 exposure. **(K)** All DESeq2 results for rhIL16 exposure for 48 hours. **(L)** All GSEA results for 48-hour rhIL16 exposure using MSigDB Hallmark gene sets. **(M)** All GSEA results for 48-hour rhIL16 exposure using GO Biological Process gene sets. **(N)** All GSEA results for 48-hour rhIL16 exposure using Reactome gene sets. **(O)** All GSEA results for 48-hour rhIL16 exposure using TE family gene sets. **(P)** Overlapping MSigDB Hallmark gene sets for *IL16* overexpression, rhIL16 24-hour exposure, and rhIL16 48-hour exposure. **(Q)** Overlapping GO Biological Process gene sets for *IL16* over-expression, rhIL16 24-hour exposure, and rhIL16 48-hour exposure. **(R)** Overlapping Reactome gene sets for *IL16* overexpression, rhIL16 24-hour exposure, and rhIL16 48-hour exposure. **(S)** Overlapping TE family gene sets for *IL16* overexpression, rhIL16 24-hour expo-sure, and rhIL16 48-hour exposure.
(XLSX)

**S8 Table. Shared gene sets that are concomitantly and significantly regulated across all conditions with upregulation of an L1 gene set. (A)** Overlapping TE family gene sets for *IL16* overexpression, *STARD5* overexpression, and rhIL16 24-hour exposure. **(B)** Overlapping MSigDB Hallmark gene sets for *IL16* overexpression, *STARD5* overexpression, and rhIL16 24-hour exposure. **(C)** Overlapping GO Biological Process gene sets for *IL16* overexpression, *STARD5* overexpression, and rhIL16 24-hour exposure. **(D)** Overlapping Reactome gene sets for *IL16* overexpression, *STARD5* overexpression, and rhIL16 24-hour exposure.
(XLSX)

**S9 Table. Age-related associations with L1 *trans*-eQTLs. (A)** Statistics for SNVs mapping to aging MeSH traits through the Open Targets Genetics platform. **(B)** Number of SNVs map-ping to unique MeSH ID disease terms. **(C)** Circulating IL16 concentrations in aging mouse serum.
(XLSX)

## Acknowledgments

We would like to thank Prof. Rachel Brem for her feedback and insights on the eQTL analyses. We would also like to thank Dr. Minhoo Kim for her feedback on the manuscript.

## Author Contributions

**Conceptualization:** Juan I. Bravo, Bérénice A. Benayoun.

**Data curation:** Juan I. Bravo, Seungsoo Kim, Bérénice A. Benayoun.

**Formal analysis:** Juan I. Bravo, Seungsoo Kim, Lucia Zhang.

**Funding acquisition:** Bérénice A. Benayoun.

**Investigation:** Juan I. Bravo, Chanelle R. Mizrahi, Seungsoo Kim.

**Methodology:** Juan I. Bravo, Seungsoo Kim, Yousin Suh, Bérénice A. Benayoun.

**Project administration:** Bérénice A. Benayoun.

**Resources:** Yousin Suh, Bérénice A. Benayoun.

**Supervision:** Juan I. Bravo, Yousin Suh, Bérénice A. Benayoun.

**Validation:** Lucia Zhang, Bérénice A. Benayoun.

**Writing – original draft:** Juan I. Bravo, Bérénice A. Benayoun.

**Writing – review & editing:** Juan I. Bravo, Chanelle R. Mizrahi, Seungsoo Kim, Lucia Zhang, Yousin Suh, Bérénice A. Benayoun.

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
