## [Decision Letter · Decision Letter 0]

13 Mar 2024

Dear Dr Benayoun,

Thank you very much for submitting your Research Article entitled 'An eQTL-based Approach Reveals Candidate Regulators of LINE-1 RNA Levels in Lymphoblastoid Cells' to PLOS Genetics.

The manuscript was fully evaluated at the editorial level and by independent peer reviewers. The reviewers appreciated the attention to an important topic but identified some concerns that we ask you address in a revised manuscript.

We therefore ask you to modify the manuscript according to the review recommendations. Your revisions should address the specific points made by each reviewer.

Yours sincerely,

Scott M. Williams

Section Editor

PLOS Genetics

Scott Williams

Section Editor

PLOS Genetics

Reviewer's Responses to Questions

**Comments to the Authors: **

Reviewer #1: I am satisfied with the novel version of the manuscript and the answers provided by the Authors. I believe the changes incorporated in the manuscript respond to the majority of the raised issues and have no additional comments on this beautiful piece of work.

Reviewer #2: Bravo et al use statistical genetics approaches to study variation in LINE-1 RNA expression in human LCLs. As evidenced by prior reviews made available, the abstract introduces LINE-1 as "autonomous, actively mobile elements" but this analysis is mostly about intronic and no longer autonomous or mobile elements transcribed as a consequence of their location in conventional gene's ranges of transcription. The introduction has been largely amended to explain this, but the abstract still points to the rare, exciting and not-exactly in view here "hot" L1s. The abstract and introduction still seemingly acknowledge that these would be the most interesting. If that is the case, I would suggest that the authors would use this same data and the same statistical tools to perform a complementary analysis: rather than look at SNVs genome wide, analyze whether the substantially fewer L1 MEVs (Kojima et al., Nat Genet) are associated with L1 RNA expression levels, or expression levels of the portion of L1 RNA that can best indicate bona fide L1 transcription from its own promoter (McKerrow et al., NAR).

As it stands, their analyses are built around an initial set of so called L1 trans-eQTLs. These would be more appropriately termed LINE1 expression QTLs, since these variants may be regulating LINE-1 RNA levels in cis- or in trans-. The point is that they looked genome wide for variants associated with LINE-1-mapping RNA levels. They found 3 true loci (plus two seemingly single-variant suggestive hits), and these are each interesting - something meaningful is going on with these loci and L1, but it is not clear to me what. They should analyze each locus in more granular structural genetic detail before adding on additional statistical genetics, e.g. mediation. They should dedicate more time to convincing readers that these 3 loci actually impact the levels of the L1 families they say they do. To me, this would require that they include PEER factors in the "trans eQTL" analyses. If they tried, and the significance evaporated, that could be informative about the nature of these loci. 

Next, they checked which cis-eQTLs "overlap" with these 499 variants. The fact they they turn up >800,000 eQTLs suggests that they looked simply at overlapping genomic ranges. They should instead have performed colocalization analyses. In the end, however, they land on index variants in LD blocks associated with both a certain L1 family expression and a certain gene's expression. The trivial explanation for such loci, as suggested by another reviewer, is that an L1 element of that family is present in the introns of the eGene, and for their strongest locus, this appears to be one possible (and to me, exceedingly plausible) explanation. The regression analysis is not a convincing addition.

The experimental work is predicated on the statistical genetic analysis, which is not yet convincing. It should be well noted that LCL gene expression is impacted substantially by B cell activation state and EBV gene expression latency program (e.g. SoRelle et al., eLife). These could likely be captured as PEER factors, but can be missed if EBV expression is. considered only as the sum of all EBV gene expression.

Reviewer #3: It is our opinion that the authors have adequately addressed reviewer concerns. In particular, the main objection raised by the reviewers having to do with a conflation between TE-promoter driven versus non-TE-promoter driven transcription was thoroughly dealt with, as is reflected in a lengthy discussion of the matter and a substantially bolstered analysis considering the genic context of TE-read origin. The revised work is much stronger for this. On the concern of whether GSEA applied to TE subfamilies is appropriate, we agree with the authors that although not commonly practiced, this is in fact the ideal tool for their stated purpose. We ask only that the authors double check whether the direction of change attributed to gene sets in figure 3 panels E-G are correct (perhaps genes were accidentally arranged in ascending Wald-statistic order?), as some of the cited literature (Wahl et al. 2020) identifies the opposite trend, namely a positive relationship between TE and mTORC1 expression. The present results are nevertheless consistent with Marasca et al. 2022 which also observe a negative relationship, but double checking should be easy enough.

Overall, we appreciate the author’s thoughtful response to the reviewer critiques. We think it makes an important contribution to the existing body of literature on the regulation of TE expression.

**Have all data underlying the figures and results presented in the manuscript been provided?** Large-scale datasets should be made available via a public repository as described in the *PLOS Genetics*
data availability policy, and numerical data that underlies graphs or summary statistics should be provided in spreadsheet form as supporting information.

Reviewer #1: Yes

Reviewer #2: Yes

Reviewer #3: Yes

PLOS authors have the option to publish the peer review history of their article (what does this mean?). If published, this will include your full peer review and any attached files.

Reviewer #1: No

Reviewer #2: **Yes: **Nicholas F. Parrish

Reviewer #3: **Yes: **John Sedivy

---

## [Editor Report · Decision Letter 1]

21 May 2024

Dear Dr Benayoun,

We are pleased to inform you that your manuscript entitled "An eQTL-based Approach Reveals Candidate Regulators of LINE-1 RNA Levels in Lymphoblastoid Cells" has been editorially accepted for publication in PLOS Genetics. Congratulations!

Yours sincerely,

Scott M. Williams

Section Editor

PLOS Genetics

Hua Tang

Section Editor

PLOS Genetics

Comments from the reviewers (if applicable):

**Data Deposition**

http://datadryad.org/submit?journalID=pgenetics&manu=PGENETICS-D-23-01425R1

**Press Queries**

---

## [Editor Report · Acceptance letter]

4 Jun 2024

PGENETICS-D-23-01425R1 

An eQTL-based approach reveals candidate regulators of LINE-1 RNA levels in lymphoblastoid cells 

Dear Dr Benayoun, 

We are pleased to inform you that your manuscript entitled "An eQTL-based approach reveals candidate regulators of LINE-1 RNA levels in lymphoblastoid cells" has been formally accepted for publication in PLOS Genetics! Your manuscript is now with our production department and you will be notified of the publication date in due course.

With kind regards,

Anita Estes

PLOS Genetics

On behalf of:
